ARTICLES

## OPEN
# The evolutionary origin of host association in the Rickettsiales

Max E. Schön [1,6], Joran Martijn[1,2,3,6], Julian Vosseberg [1,4], Stephan Köstlbacher [5] and
Thijs J. G. Ettema [1,5] ✉

The evolution of obligate host-association of bacterial symbionts and pathogens remains poorly understood. The Rickettsiales are an alphaproteobacterial order of obligate endosymbionts and parasites that infect a wide variety of eukaryotic hosts, including humans, livestock, insects and protists. Induced by their host-associated lifestyle, Rickettsiales genomes have undergone reductive evolution, leading to small, AT-rich genomes with limited metabolic capacities. Here we uncover eleven deep-branching alphaproteobacterial metagenome assembled genomes from aquatic environments, including data from the *Tara* Oceans initiative and other publicly available datasets, distributed over three previously undescribed Rickettsiales-related clades. Phylogenomic analyses reveal that two of these clades, Mitibacteraceae and Athabascaceae, branch sister to all previously sampled Rickettsiales. The third clade, Gamibacteraceae, branch sister to the recently identified ectosymbiotic 'Candidatus Deianiraea vastatrix'. Comparative analyses indicate that the gene complement of Mitibacteraceae and Athabascaceae is reminiscent of that of free-living and biofilm-associated bacteria. Ancestral genome content reconstruction across the Rickettsiales species tree further suggests that the evolution of host association in Rickettsiales was a gradual process that may have involved the repurposing of a type IV secretion system.

Obligate host-associated bacteria include pathogens that represent a leading cause of human, livestock and crop disease, resulting in considerable economic loss worldwide. While the molecular and cellular underpinnings of host association have been described in considerable detail[1], their evolutionary origin remains generally poorly understood. The Rickettsiales represent a widespread and diverse order of obligate host-associated alphaproteobacteria that have been estimated to have originated >1.7 Ga, similar to a conservative estimated age of eukaryotes[2,3]. Rickettsiales infect a wide variety of eukaryotic species, including protists, leeches, cnidarians, arthropods and mammals[1]. Well-known examples include *Rickettsia prowazekii*, the causative agent of epidemic typhus in humans[4], and *Wolbachia*, a genus of bacteria infecting over two-thirds of arthropods and nearly all filarial nematodes[5]. Genomes of Rickettsiales are shaped by ongoing reductive evolution and are typically small (<1.5 Mb), rich in A+T nucleotides (<40% G+C), display a low coding density (<85%), lack metabolite biosynthesis genes and display a high degree of pseudogenization[6,7]. The host's nutrient-rich cytoplasm rendered biosynthetic genes redundant, and genetic drift enhanced by small effective population sizes and frequent bottlenecks resulted in further genomic deterioration[8]. Rickettsiales employ various host-interaction factors, including a characteristic P-type (or Rickettsiales *vir* homologue, *rvh*[9]) type IV secretion system (T4SS), host-cell manipulating effector proteins[1,10–12] and an ATP/ADP translocase. The latter facilitates energy parasitism by exchanging host-cell ATP for endogenous ADP[13,14] and is commonly found in host-associated bacteria[15]. The currently recognized Rickettsiales families (Rickettsiaceae, Anaplasmataceae, Midichloriaceae and Deianiraeaceae) have each adopted specific lifestyles to interact with their respective host-cell environment[1,16–19]. While these

observations give us insights into how the Rickettsiales adapted to the intracellular environment, it is still unclear how and when their last free-living ancestor became host-associated initially.

Here we describe the discovery of genomes of deeply branching rickettsial lineages. In-depth analyses of these genomes suggest that their lifestyle is reminiscent of that of free-living and biofilm-colonizing planktonic bacteria. Subsequent ancestral genome content analysis across Rickettsiales provides new insights about the emergence of host association, a key step in the evolution of various host relationships displayed by this bacterial clade, including pathogenicity, mutualism and reproductive parasitism.

## Results

**Metagenomic identification of previously undescribed Rickettsiales.** To shed light on the early evolution of Rickettsiales and the emergence of host association within this clade, we screened publicly available metagenomic repositories for deep-branching Rickettsiales (Supplementary Data 1 and 2). We reconstructed three metagenome assembled genomes (MAGs) from the *Tara* Oceans metagenome data[20] and identified another eight MAGs from public data derived from aquatic (marine, lake, aquifer and tailings water) environments (Supplementary Fig. 1 and Data 3)[21–24]. To assess their phylogenetic affiliation with other alphaproteobacteria and mitochondria, we inserted them into a previously established dataset of 24 genes highly conserved in alphaproteobacterial and gene-rich mitochondrial genomes[25]. We found that the identified lineages represented three distinct, previously undescribed Rickettsiales clades unrelated to mitochondria (Fig. 1 and Extended Data Fig. 1).

**Phylogenomic placement of obtained clades.** To pinpoint the phylogenetic position of the retrieved Rickettsiales-associated clades

[1]Department of Cell and Molecular Biology, Science for Life Laboratory, Uppsala University, Uppsala, Sweden. [2]Department of Medical Biochemistry and Microbiology, Uppsala University, Uppsala, Sweden. [3]Department of Biochemistry and Molecular Biology, Dalhousie University, Halifax, Canada. [4]Theoretical Biology and Bioinformatics, Department of Biology, Utrecht University, Utrecht, The Netherlands. [5]Laboratory of Microbiology, Wageningen University and Research, Wageningen, The Netherlands. [6]These authors contributed equally: Max E. Schön, Joran Martijn. ✉e-mail: thijs.ettema@wur.nl

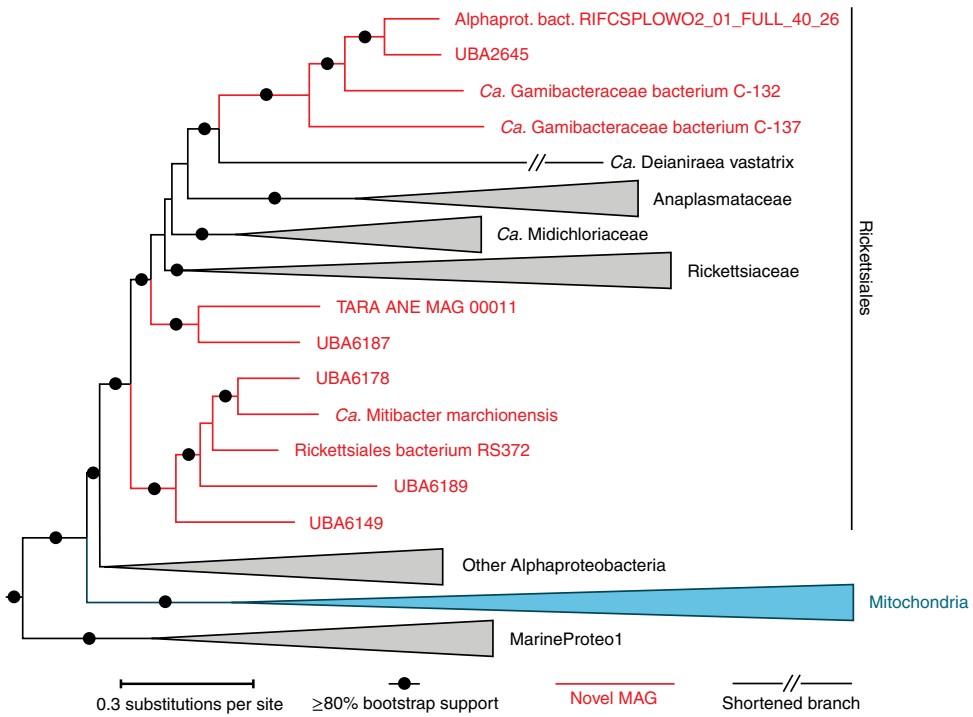

**Fig. 1 | Identification and phylogenomics of previously undescribed Rickettsiales-associated alphaproteobacteria.** Phylogenetic tree based on the 24 'alphamitoCOGs' dataset including the Rickettsiales MAGs and other recently sequenced genomes. The 20% most heterogeneous sites were removed before tree inference with IQTREE under the PMSF approximation of the LG+C60+F+Γ4 model and 100 non-parametric bootstraps. The tree was rooted with Beta-, Gammaproteobacteria and Magnetococcales. See Extended Data Fig. 1 for the uncollapsed tree.

more confidently, we performed in-depth phylogenomic analyses with a dataset comprising 116 marker genes highly conserved among Rickettsiales[26] (Supplementary Fig. 2). These analyses confirmed that two of the three clades (hereafter Mitibacteraceae and Athabascaceae; Supplementary Text) branched as distinct groups sister to all other sampled Rickettsiales (Supplementary Figs. 3 and 4, and Data 1). Of the two, the Mitibacteraceae represented the deepest branching clade. The third clade (hereafter Gamibacteraceae; Supplementary Text) branched sister to the recently described 'Candidatus Deianiraea vastatrix' (Fig. 2a), a host-associated bacterium able to replicate outside their host cells[17]. The placement of the clade comprising Gamibacteraceae and Deianiraeaceae within the Rickettsiales species tree was not fully resolved. In agreement with Castelli et al.[17], this clade branched sister to the Anaplasmataceae with maximum branch support in both Bayesian (Supplementary Fig. 3) and maximum likelihood (Supplementary Fig. 4) trees. However, alphaproteobacterial and possibly rickettsial phylogenies[25,27–29] are known to be affected by long branch and compositional bias artefacts. We tested for such artefacts by separately removing the long-branched 'Ca. D. vastatrix' and the most heterogeneous sites (Methods and Supplementary Text). The overall topology was robust to either treatment (Supplementary Figs. 5 and 6), except for the placement of the Gamibacteraceae-Deianiraeaceae clade. When removing the most heterogeneous sites, they branched sister to a clade comprising the Midichloriaceae and Anaplasmataceae with near maximum branch support in the Bayesian tree (Fig. 2a, Supplementary Fig. 7 and Table 1) but with insignificant branch support in the maximum likelihood tree (Supplementary Text and Fig. 8). On the basis of the resolved Bayesian phylogeny, we suggest that this topology more probably reflects Rickettsiales evolutionary history and that the placement of the Gamibacteraceae-Deianiraeaceae clade sister to Anaplasmataceae is the result of a phylogenetic artefact.

**Environmental distribution of added Rickettsiales.** To assess the environmental distribution of the newly obtained alphaproteobacterial clades, we used the 16S ribosomal RNA gene sequences of Gamibacteraceae and Mitibacteraceae MAGs to query public sequence databases. Athabascaceae MAGs did not contain 16S rRNA genes and could therefore not be included in this analysis. Highly similar sequences were exclusively found in datasets obtained from aquatic habitats. Sequences closely related to the Gamibacteraceae 16S rRNA gene were associated with a diverse set of environments, including lakes and aquifers as well as marine systems. Interestingly, two recovered 16S rRNA gene sequences were obtained from sequencing datasets of cells of the freshwater cnidarian *Hydra vulgaris* and of the marine ciliate *Hemigastrostyla elongata* (Fig. 2b and Supplementary Fig. 9)[30,31]. The two sequences are unlikely to be contaminants as rigorous efforts were made to minimize contamination[30,31]. Their detection in a ciliate microbiome could hint at a conserved lifestyle of Gamibacteraceae and their closest characterized relative 'Ca. D. vastatrix', which was described to colonize the cell surface of its ciliate host *Paramecium primaurelia*[17]. Apart from Gamibacteraceae, Deianiraeaceae-related 16S rRNA gene sequences were also detected in the *Hydra vulgaris* microbiome (Supplementary Fig. 9)[17,30]. Most probably, the observed interactions of Gamibacteraceae and Deianiraeaceae with *Hydra* can be explained by the presence of host ciliates in the polyp microbiome. Alternatively, *Hydra* could represent a direct host for these lineages as well. We could not detect potential host organisms for Mitibacteraceae, as all 'Ca. Mitibacter marchionensis'-related 16S rRNA gene sequences were exclusively associated with marine, non-host-associated environments (Fig. 2c and Supplementary Fig. 9).

**Inferred physiology of Mitibacteraceae and Athabascaceae.** The reconstructed genomes of Mitibacteraceae and Athabascaceae are on average larger in size, display a higher GC content and a

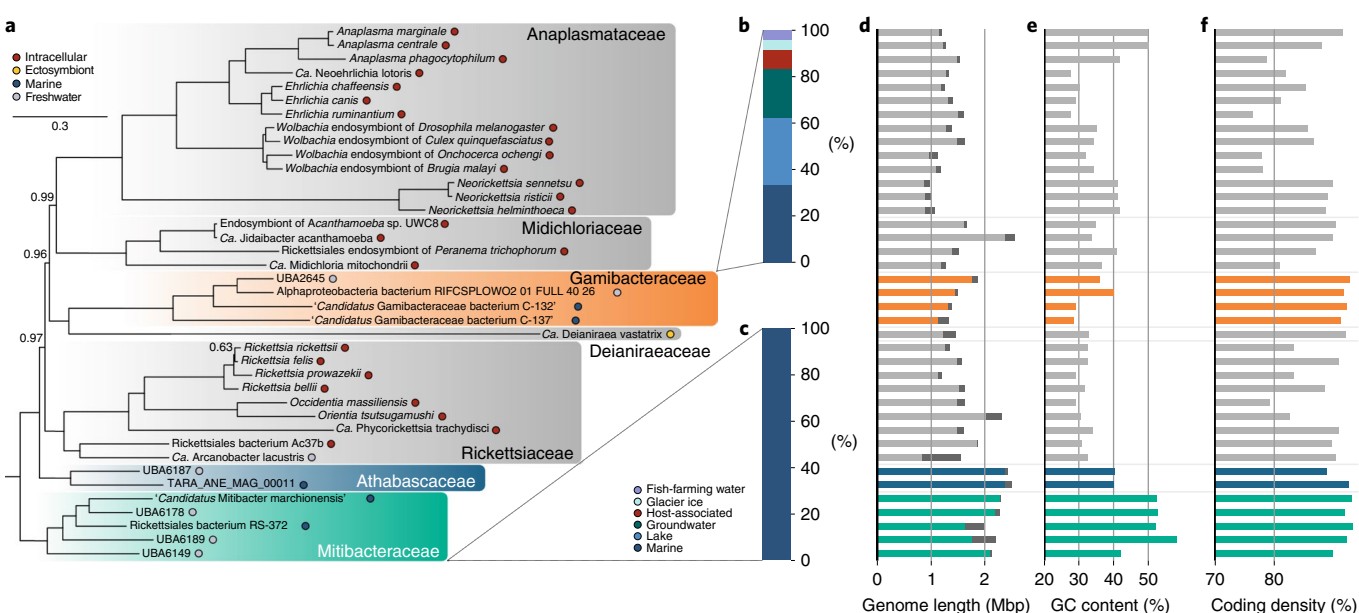

**Fig. 2 | Phylogeny and comparative analysis of deep-branching Rickettsiales genomes. a**, Rickettsiales species tree based on a dataset of 116 marker genes of which the most heterogeneous sites were removed, highlighting the phylogenetic position of Gamibacteraceae (orange shading), Athabascaceae (blue shading) and Mitibacteraceae (green shading). The tree was inferred using PhyloBayes (CAT+LG+Γ4, 20,000 generations with a burn-in of 5,000 generations). Support values correspond to posterior probabilities and are only shown for values below 1. See Supplementary Fig. 7 for the complete tree including the outgroup. **b–f**, Distribution of environmental Gamibacteraceae (**b**) and Mitibacteraceae sequences (**c**) obtained from the NCBI nt database. Colours represent the sequences' source environments. For each genome, the observed genome size in Mbp (**d**), with estimated genome size in dark grey shading, the GC content (**e**) and the coding density, calculated as the total length of protein-coding genes divided by the genome size (**f**), are shown. Source data for graphs in **d–f** can be found in Supplementary Data 3.

higher protein-coding density (Fig. 2d–f and Supplementary Data 3) compared with those of the known obligate host-associated Rickettsiales (hereafter referred to as 'classical Rickettsiales', including Gamibacteraceae). Mitibacteraceae and Athabascaceae, like most classical Rickettsiales, encode an incomplete glycolysis pathway and a complete tricarboxylic acid cycle (Extended Data Fig. 2 and Supplementary Data 4). However, unlike all classical Rickettsiales, they encode complete pathways to synthesize all amino acids and nucleotides (Fig. 3a, Extended Data Fig. 2 and Supplementary Data 4). Even the ectosymbiotic 'Ca. D. vastatrix', by far the richest classical Rickettsiales described so far in terms of amino acid biosynthesis potential, can only synthesize 15 amino acids and lacks the genetic potential for de novo nucleotide biosynthesis (Fig. 3a, Extended Data Fig. 2 and Supplementary Data 4)[17]. The two lineages further encode nearly all subunits of the main oxidative phosphorylation complexes and a complete glyoxylate cycle. The latter may allow them to use substrates such as acetate as a sole carbon source (Extended Data Fig. 2 and Supplementary Data 4). They further uncharacteristically encode functions more commonly found in free-living bacteria, such as a sulfate uptake system and a complete (Athabascaceae) or partial (Mitibacteraceae) assimilatory sulfate reduction pathway, with the potential to provide sulfide for the de novo biosynthesis of cysteine (Extended Data Fig. 2 and Supplementary Data 4)[32,33]. Two Mitibacteraceae and one Athabascaceae MAG also feature *ars* gene clusters (Supplementary Data 4), which are lacking in all classical Rickettsiales. These clusters facilitate arsenite export (Extended Data Fig. 2) and so confer arsenic resistance. Arsenic is the most prevalent toxic element in the environment[34] and resistance mechanisms are almost ubiquitously found in prokaryotes[35]. Mitibacteraceae further encode an ammonium transporter which is an important inorganic nutrient uptake system in marine bacteria and is not found in other Rickettsiales[36]. Despite being aquatic alphaproteobacteria, the Mitibacteraceae

and Athabascaceae do not encode proteorhodopsin-related proteins (Supplementary Data 4), indicating that they are unable to harness energy from a light-driven H+ gradient. However, they do encode homologues of hydroperoxidase KatG (not found in other Rickettsiales families) and superoxide dismutase SOD2 (Extended Data Fig. 2, Supplementary Data 4 and 6), suggesting that they experience stress from UV light-induced reactive oxygen species.

Furthermore, we found that Mitibacteraceae and Athabascaceae genomes encode a complete flagellum and associated chemotaxis machinery (Fig. 3a), suggesting a possible motile lifestyle[17,18]. Their genomes also encode Type 4 pili (T4P; Fig. 3a), which have been implicated in motility, surface attachment and biofilm formation[37], but can also function in evasion of protist predators[38]. Mitibacteraceae genomes additionally encoded a tight adherence pilus (Tad) and an extracellular polysaccharide biosynthesis pathway via *pel* gene clusters[39,40]. The latter was also found in the Athabascaceae MAG UBA6187 (Fig. 3a and Extended Data Fig. 3a,b). Both Tad pili and exopolysaccharide biosynthesis capacity have been shown to play important roles in biofilm formation[41,42] and together with the presence of T4P, suggest that Mitibacteraceae, and perhaps Athabascaceae, can form biofilms.

We failed to detect genes encoding an ATP/ADP translocase or any other gene typical of intracellular or parasitic lifestyles in Mitibacteraceae and Athabascaceae, except for those encoding the *rvh* T4SS, including the duplication of several components[43] as well as several distinct modifications compared with the *vir* T4SS of *Agrobacterium*[10] (Fig. 3b,c and Supplementary Fig. 10). In classical Rickettsiales, this T4SS is typically used for the manipulation of the host cell via translocation of effector proteins[10–12].

Besides a few homologues of rickettsial ankyrin repeat protein 1 (RARP-1), a Sec-TolC-secreted effector of *Rickettsia typhi*[44], and RARP-2 which may be secreted by the T4SS, we were neither able to identify any of the experimentally verified T4SS effector

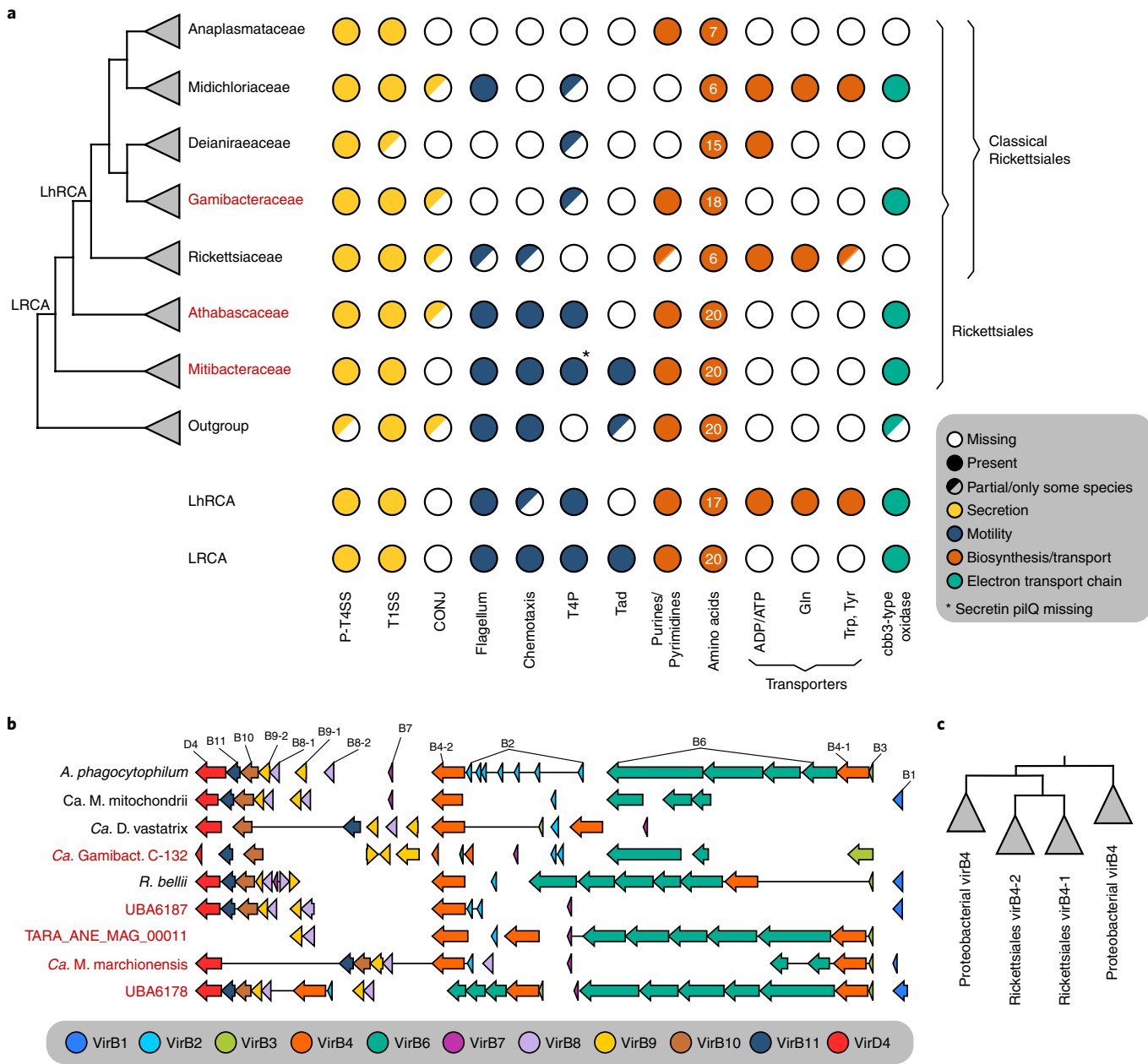

**Fig. 3 | Distribution and ancestral inference of phenotypic traits and pathways across Rickettsiales. a**, Presence (full circle) and absence (empty circle) of genes or pathways in the seven Rickettsiales families (including Gamibacteraceae, Athabascaceae and Mitibacteraceae), the alphaproteobacterial outgroup as well as the last common ancestors of the classical, obligate host-associated Rickettsiales (LhRCA) and of all Rickettsiales (LRCA). Pathways are grouped into 'secretion' (yellow), 'motility' (blue), 'biosynthesis and transport' (orange) and 'electron transport chain' (green). In some cases, pathways are either partial (one or several genes missing) or are absent from some species, but present in others (half circle). Numbers inside the circles represent the number of complete pathways for the biosynthesis of amino acids. The Gln transporter could potentially also transport other polar amino acids such as Ser, Thr and Asn. **b**, Synteny of the Rickettsiales *vir* homologue (*rvh*) T4SS in classical (black) and environmental (red) Rickettsiales. Taxa are ordered according to the species tree in **a**. One representative per family except Athabascaceae and Mitibacteraceae with two representatives each. **c**, Schematic phylogenetic tree showing the common duplication of the *vir*B4 gene in LRCA. See Supplementary Fig. 10 for the uncollapsed tree. P-T4SS, type IV secretion system of the P type; T1SS, type I secretion system; CONJ, conjugation system; T4P, type 4 pilus; Tad, tight adherence system.

proteins of classical Rickettsiales[11,12], nor could we detect similar numbers of putative effector proteins containing eukaryotic-like repeat domains as in other Rickettsiales genomes (ankyrin-, leucine rich- and tetratricopeptide repeats; Supplementary Text, Extended Data Fig. 4 and Supplementary Data 4). In contrast, we detected several proteins containing peptidoglycan-binding (PGB) domains in Mitibacteraceae genomes. These proteins, which are otherwise relatively rare in Rickettsiales (Supplementary Text and Extended

Data Fig. 4), have been shown to serve as effectors of a 'bacteria killing'-type T4SS in other proteobacteria[45,46]. However, PGB domain proteins are also known to operate in peptidoglycan synthesis and cell shape remodelling[47], and may thus allow the Mitibacteraceae to evade predatory protists by providing cell shape plasticity[48].

Based on their general genome structure and gene content, Mitibacteraceae and Athabascaceae are atypical compared with the classical obligate intracellular Rickettsiales. Instead, they possibly

exhibit a lifestyle similar to aquatic copiotrophic, biofilm-associated and free-living bacteria[49–51], or perhaps a facultative host-associated (probably extracellular) lifestyle (Figs. 2d–f and 3a, Extended Data Fig. 2 and Supplementary Data 4). The suggestion that Mitibacteraceae and Athabascaceae are possibly free-living bacteria is reinforced by an analysis with PhenDB[52], a machine learning algorithm that has been trained to predict phenotypic traits, including obligate host-association, on the basis of gene content. PhenDB predicted that Mitibacteraceae and Athabascaceae are not obligate intracellular symbionts, in contrast to classical Rickettsiales (Supplementary Table 2).

**Emergence of host association in Rickettsiales.** The deep-branching nature of the Mitibacteraceae and Athabascaceae allowed us to study the origin of the host-associated lifestyle of the Rickettsiales in unprecedented detail. We reconciled[53,54] 4,240 single-gene trees with the previously obtained species tree (Fig. 2a) to reconstruct the gene family complement as well as gene duplications, transfers, losses and origination events along the entirety of the species tree (Extended Data Fig. 5, Supplementary Fig. 11, and Data 4 and 5). Below, we focus on two ancestors that are central to understanding the emergence of host association: the last Rickettsiales common ancestor (LRCA) and the last obligate host-associated Rickettsiales common ancestor (LhRCA; Fig. 3a).

The LRCA was inferred to feature at least 1,432 protein-coding genes, considerably more than most extant classical Rickettsiales (Supplementary Data 6). This is most probably an underestimate as LRCA gene families could have gone extinct or were not sampled in the analysed genomes. Only a few classical Rickettsiales encode a similar number or more protein-coding genes. For example, *Orientia tsutsugamushi* str. Boryong encodes 2,120 protein-coding genes, but this large number can be explained by a recent massive proliferation of transposons, conjugative T4SS genes and other host-cell interaction genes[55]. Furthermore, this expanded gene repertoire of *O. tsutsugamushi* also includes around 800 pseudogenes, most of which are duplicates of functional genes[55,56]. The *rvh* T4SS in LRCA was most probably acquired via horizontal gene transfer from an ancestral proteobacterial donor (Fig. 3c and Supplementary Fig. 10). We were unable to assign any verified Rickettsiales effector proteins[1,11,12] related to eukaryotic host-association to this T4SS, suggesting that it could have served an alternative function. As we inferred LRCA to contain at least four PGB domain-containing proteins, which are putative effectors of such secretion systems, it could have acted instead as a 'bacteria killing' type T4SS[45,46]. Alternatively, the ancestral T4SS could also have acted in defence against predatory protists. The LRCA was further inferred to encode a very similar gene profile as observed for the Mitibacteraceae and Athabascaceae, hence including many genes affiliated to a free-living lifestyle (for example, full complement of nucleotide and amino acid biosynthesis genes; Fig. 3a, Extended Data Fig. 2 and Supplementary Data 4) and lacking several genes related to the typical symbiotic lifestyle of classical Rickettsiales (for example, an ATP/ADP translocase and several conserved amino acid transporters; Fig. 3a). Besides the capacity to synthesize its own nucleotides and amino acids, LRCA had the genetic potential to take up sulfate and ammonium, carry out assimilatory sulfate reduction and confer arsenic resistance (Fig. 4, Extended Data Fig. 2 and Supplementary Data 4). It had a motile lifestyle, enabled by a flagellum, a T4P and a functioning chemotaxis system. Finally, it most probably was able to initiate biofilm formation using the *pel* and *tad* systems (Fig. 4 and Supplementary Data 4). On the basis of these observations, we propose that LRCA was not an obligate intracellular symbiont, but rather a free-living or perhaps facultative host-associated organism. This would avoid the need for an unprecedented and highly complex intracellular-to-extracellular transition to explain the extracellular lifestyle of '*Ca.* D. vastatrix', as originally argued by Castelli et al.[17]. However, a stage of facultative intracellular host-association cannot be fully excluded either.

The transition from the LRCA to the last common ancestor of all classical Rickettsiales and the Athabascaceae (LhRAtCA) saw little net change to the number of inferred protein-coding genes (from $n = 1,432$ in the LRCA to $n = 1,430$ in LhRAtCA). Besides the loss of several genes encoding components of the Tad pilus in LhRAtCA, no genes of note were lost or gained (Fig. 3a). In contrast, the LhRCA had considerably fewer genes ($n = 1,165$) and had lost genes involved in alanine, cysteine and serine biosynthesis, sulfate assimilation, sulfate transporters and ammonium transporters (Fig. 4). It furthermore lost the *pel* genes, suggesting that the ability to form biofilms was lost at this stage (Figs. 3a and 4). The characteristic ATP/ADP translocase which enables energy parasitism was gained here, as well as a transporter for polar amino acids, such as glutamine, and a tyrosine or tryptophan transporter. Thus, the LhRCA, as its extant descendants, most probably relied on a host for certain key metabolites but was also still able to synthesize a wide range of metabolites itself (Fig. 4 and Supplementary Data 4). We therefore hypothesize that the LhRCA was the first Rickettsiales to be an obligate symbiont. It did retain all flagellum and T4P genes, many of the chemotaxis-related genes and the ability to synthesize nucleotides de novo. Taken together, we suggest that it exhibited an obligate symbiotic yet extracellular lifestyle, which was conserved in its descendant '*Ca.* D. vastatrix'[17], and possibly also in the Gamibacteraceae. However, based on current data, a facultative intracellular lifestyle cannot be excluded either. Its host was most probably a unicellular eukaryote, as the LhRCA was previously estimated, via molecular dating, to predate the origin of Metazoa by ~700 Myr and predicted, via ancestral trait reconstruction, to be protist-associated[2]. Irrespectively, all known extant descendants of the LhRCA retained this obligate symbiotic lifestyle, but diversified to occupy distinct ecological niches.

**Key transitions in classical Rickettsiales.** From the LhRCA onwards, the evolutionary history of the classical Rickettsiales is generally characterized by reduction of central metabolic capabilities (Extended Data Figs. 2 and 5, and Supplementary Data 6). This trend is typical for obligate host-associated clades[8,19,57]. Yet, a number of genes linked to host association were gained as well. Below we discuss several observations that are of particular interest with respect to the evolution of host association.

Assuming an extracellular nature of the LhRCA, one of the key transitions that occurred during Rickettsiales evolution was the emergence of an intracellular lifestyle. Castelli et al.[17] proposed that intracellularity in Rickettsiales evolved multiple times independently. Given our species tree (Fig. 2a), this would entail independent transitions to intracellularity in the last common ancestor of Midichloriaceae and Anaplasmataceae (LMiACA), and in the last common ancestor of the Rickettsiaceae (LRiCA). The extensive loss of a largely overlapping set of amino acid biosynthesis genes coincided with these transitions (Extended Data Fig. 5): the LMiACA lost the capacity to synthesize nine amino acids, and the LRiCA eleven amino acids. In contrast, the possibly ectosymbiotic Gamibacteraceae retained its amino acid biosynthesis pathways and the ectosymbiont '*Ca.* D. vastatrix' only lost three such pathways. This suggests that switching to a predominantly intracellular environment predisposed these lineages to losing many amino acid biosynthesis genes.

Similar to the emergence of an intracellular lifestyle, the association with animal hosts evolved independently in several lineages of classical Rickettsiales. Like the LhRCA, which was inferred to be a protist symbiont[2,17], species from the Rickettsiaceae and Midichloriaceae and '*Ca.* D. vastatrix' are known to be associated with protists such as ciliates and amoeba. Wang and Luo[2] inferred that the highly specialized insects and mammal symbionts

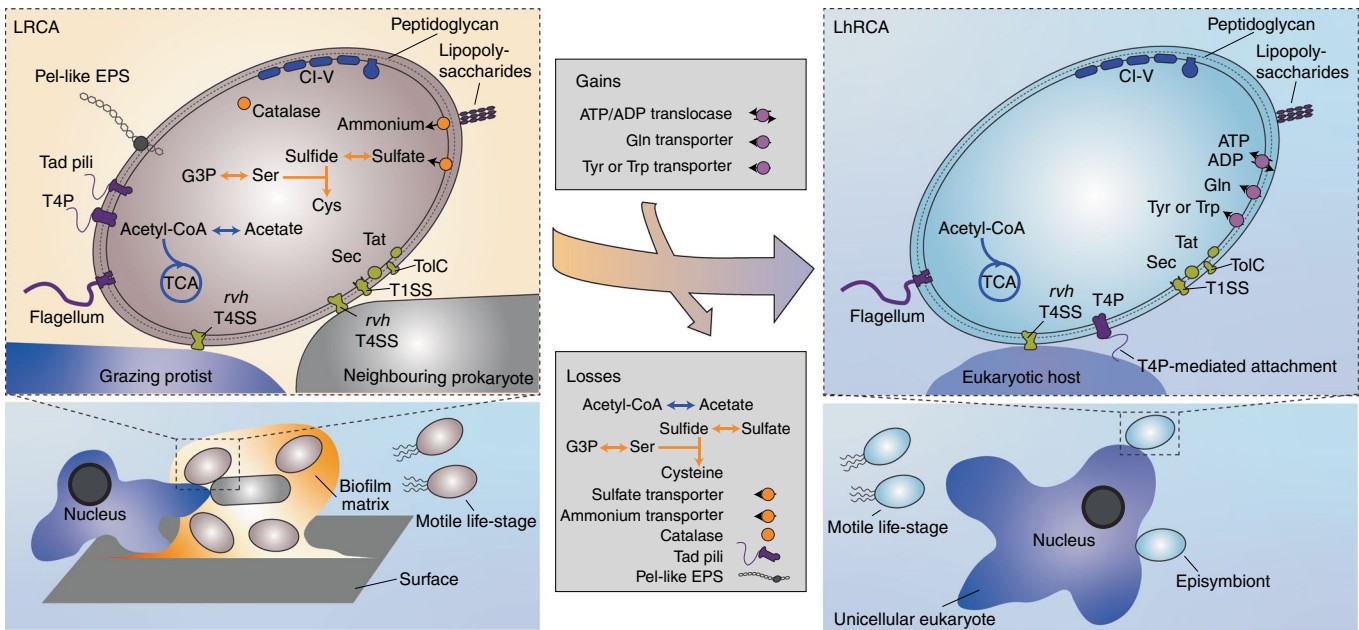

**Fig. 4 | Evolutionary transition from a free-living to a host-associated lifestyle in Rickettsiales ancestors.** Proposed lifestyle of LRCA and LhRCA in blue boxes at the bottom indicating marine environments. Rickettsiales cells are depicted as ellipsoid cell models in brown and blue for LRCA and LhRCA, respectively. Reconstructions of key genomic features of ancestors are depicted in cell models in the respective boxes. Arrow in the centre indicates evolutionary transition from LRCA to LhRCA and grey boxes above and below show gains and losses of genomic traits, respectively. Orange and blue arrows indicate metabolic pathways related to sulfate assimilation or central carbon metabolism, respectively. Abbreviations in the figure refer to genes encoding for the tight adherence (Tad) pili; type 4 pili (T4P); type I (T1SS) or *rvh* type IV (T4SS) secretion systems; Pel-like exopolysaccharide (EPS) synthesis; tricarboxylic acid (TCA) cycle; electron transport chain complexes (C) I-V; Sec-, Tat- and TolC-mediated protein secretion pathways (Sec, Tat and TolC, respectively).

and pathogens of the Anaplasmataceae, Midichloriaceae and Rickettsiaceae evolved by frequent and independent transitions from protist-associated ancestors[2,17]. A similar host switch could have occured in Gamibacteraceae as well (see above).

The hallmark feature of energy parasitism, the ATP/ADP translocase[13–15], was lost twice independently as well: once in the last common ancestor of the Gamibacteraceae (LGCA) and once in the last common ancestor of the Anaplasmataceae (LACA). The Anaplasmataceae can produce their own ATP via oxidative phosphorylation using the electron transport chain[1] and the Gamibacteraceae have the genetic potential for this as well (Extended Data Fig. 2). The lack of an ATP/ADP translocase in Gamibacteraceae suggests that this group may be less dependent on a host for energy and nutrients (Fig. 3a and Extended Data Fig. 2).

The genetic potential for storing carbon as polyhydroxybutyrate (PHB) was lost twice independently as well: once in the last common ancestor of the Gamibacteraceae and 'Ca. D. vastatrix' (LGDCA), and once in the LACA (Extended Data Figs. 2 and 5, and Supplementary Data 4). Rickettsial species may store carbon as PHB when host energy sources are unavailable[7], for instance, when they are horizontally transmitted to a new host. PHB carbon storage is otherwise commonly found in marine heterotrophic bacteria[58]. Intriguingly, losing PHB carbon storage coincided in both cases with the loss of the flagellum (Extended Data Fig. 5) and may indicate a diminished capability for both ancestors to move and survive independent of a host.

DsbB, a protein previously proposed to be involved in oxidative folding of secreted *Wolbachia* effectors[59], was gained by the LMiACA, as was a DnaJ-like chaperone. Genomes of the endosymbiotic Midichloriaceae and Anaplasmataceae experience an increased deleterious mutational load due to their intracellular nature[8,60]. This chaperone could potentially help stabilize enzymes to retain their function despite an accumulation of destabilizing mutations in their

respective genes[60,61]. The LRiCA additionally gained several genes involved in host entry and manipulation, including the known family of autotransporters called surface cell antigens (Sca)[4,62], the known Sec-TolC-secreted effector RARP-1[44] and several copies of related ankyrin repeat-containing proteins (Extended Data Fig. 5 and Supplementary Data 4).

FhaB and FhaC, proteins involved in adhesin production and export[63], were gained by the LGDCA (Extended Data Fig. 5 and Supplementary Data 4). The homologues in the ectosymbiont 'Ca. D. vastatrix' (DeiVas_00243/WP_146820351.1 and DeiVas_00245/WP_161982782.1) were previously proposed to be involved in adhesion to cells of the host *Paramecium primaurelia*[17], as they are essential for host-cell adhesion in *Bordetella* species as well[63]. Their conservation in 'Ca. D. vastatrix' and Gamibacteraceae MAGs underpins a probably ancestral ectosymbiotic lifestyle of this group.

## Discussion

The origin and emergence of host association during the early evolution of Rickettsiales remains poorly understood. Here we identify and reconstruct eleven MAGs distributed over the proposed deep-branching Mitibacteraceae and Athabascaceae and the more nested Gamibacteraceae. We propose that the Mitibacteraceae and Athabascaceae exhibit free-living or possibly facultative host-associated lifestyles, and are capable of forming biofilms. Their genomes lack the signatures of reductive evolution and contain the genetic repertoire for chemotactic motility, biofilm formation and, most importantly, a rich metabolism with complete biosynthetic pathways for all amino acids and nucleotides. Subsequent ancestral genome reconstruction analyses indicate a similar, free-living lifestyle for the LRCA, although a facultative host-association cannot be fully excluded. The *rvh*-type T4SS, characteristic for Rickettsiales, was acquired by the LRCA through horizontal gene transfer (HGT) from a proteobacterial donor. This ancestral T4SS may have functioned in so-called 'bacteria

killing'[45,46] as indicated by the presence of several PGB-containing proteins in the LRCA, and/or in defence against protist predation. Evasion of predatory protists may have additionally been facilitated by the aforementioned PGB proteins, which are known to function in cell shape remodelling[47], and by biofilm formation, which was shown to allow some aquatic prokaryotes to evade protist grazing[48,64]. As the Rickettsiales evolved into obligate symbionts of ancestral protists, we propose that the *rvh* T4SS was repurposed from a system for inter-bacterial competition ('bacteria killing') and/or defence against pro-tist predation into the host-cell manipulating secretion system found in present-day Rickettsiales symbionts. In addition to the proposed repurposing of the T4SS, the loss of biofilm-forming capacity and key metabolic genes, mirrored by the gain of several amino acid transport-ers and an ATP/ADP translocase in the LhRCA, represents key events in the transition from a free-living or facultative host-associated life-style to an obligate symbiotic lifestyle during Rickettsiales evolution. We propose the LhRCA to have been an ectosymbiont of an ancestral protist, a lifestyle that is conserved in '*Ca.* D. vastatrix' and possibly also in Gamibacteraceae. Alternatively, it could have exhibited a facul-tative intracellular lifestyle, and its known descendants either reverted to extracellularity or became obligately intracellular. The LhRCA pos-sibly used its vertically inherited T4P to attach to its host, perhaps in a fashion similar to how the predatory bacterium *Vampirococcus lugosii* attaches to its prey[65]. It may further have used its flagellum and chemotaxis machinery, conserved in some extant members of the Midichloriaceae and Rickettsiaceae[16,26], for motility and to navigate towards host cells. Our results show that subsequent rickettsial evolu-tion was dominated by further genome reduction and specialization towards different host organisms and niches within or outside host cells, giving rise to the diverse lifestyles displayed by extant rickettsial symbionts and pathogens.

The current identification of previously undescribed environ-mental clades of Rickettsiales has allowed us to reconstruct the early evolution and emergence of host association in this bacterial clade. While the inferred genome content of the LRCA is compat-ible with a free-living lifestyle, we currently cannot exclude the pos-sibility that it displayed a facultative extracellular or even intracellular host-association. As our ancestral genome reconstruction analyses indicate the inferred genomic content of the LRCA to be largely con-gruent with that of present-day representatives of the Mitibacteraceae and Athabascaceae, future studies aiming to characterize the physiol-ogy and lifestyles of these lineages might provide further insights into the nature of the last common ancestor of Rickettsiales.

## Methods

**Sample selection.** All publicly available *Tara* Oceans assemblies[20] were screened with the RP15 pipeline[66,67] for the presence of Rickettsiales-related lineages, as previously described[25]. The RP15 pipeline approximates the phylogenetic position of all taxa present in a metagenome assembly for which at least 5 out of 15 well-conserved ribosomal proteins are encoded on a single contig. In the end, two assemblies (125_MIX_0.22-3 and 067_SRF_0.22-0.45) were identified (Supplementary Data 2).

**Raw sequence data.** The raw sequence data from all samples corresponding to the two selected assemblies and from additional samples (Supplementary Data 2) were downloaded from the *Tara* Oceans project ERP001736 on the EBI Metagenomics portal.

**Read preprocessing.** All reads were preprocessed as previously described[25]. SEQPREP v1.3.2 (https://github.com/jstjohn/SeqPrep) was used to merge overlapping read pairs into single reads and remove read-through Illumina adapters. TRIMMOMATIC v0.35[68] was used to remove residual Illumina adapters, trim low-quality base-calls at starts and ends of reads, remove short reads and finally remove reads that had a low average phred score. The overall quality and presence of adapter sequences of processed and unprocessed reads were assessed with FASTQC v0.11.4[69].

**Metagenome assembly.** The preprocessed metagenomic reads from the two selected samples (125_MIX_0.22-3 and 067_SRF_0.22-0.45) were re-assembled with metaSPAdes[70], a mode of SPAdes 3.7.0 with k-mers 21,33,55,77. In case a

sample was associated with multiple sequencing runs, all preprocessed reads from the different sequencing runs were pooled before assembly.

**Phylogenetic diversity in metagenome assemblies and public MAG datasets.** The RP15 pipeline was used to estimate the phylogenetic diversity of alphaproteobacterial lineages present in the two re-assembled metagenomes and published MAGs of Parks et al.[23] (PRJNA348753), Tully et al.[24] (PRJNA391943), Delmont et al.[22] (https://doi.org/10.6084/m9.figshare.4902923) and Anantharaman et al.[21] (PRJNA288027). Only the MAGs categorized by the aforementioned studies as 'Alphaproteobacteria' or 'Rickettsiales' were considered. Protein-coding sequences were predicted with Prodigal v2.60[71]. The RP15 pipeline was first run with a reference set of 90 representative bacteria and archaea to identify alphaproteobacterial contigs (Supplementary Data 1)[25]. The ribosomal proteins encoded on these contigs were then incorporated in a second RP15 dataset consisting of their orthologues in 84 representative alphaproteobacteria, 12 mitochondria, 2 MarineProteo1, 2 magnetococcales, 4 betaproteobacteria and 4 gammaproteobacteria. A concatenated supermatrix alignment was prepared (alignment: MAFFT L-INS-i v7.471, alignment trimming: trimAl v1.4.rev15 -gt 0.5). To reduce compositional bias—a phylogenetic artefact to which alphaproteobacteria are particularly sensitive[25,27,28,72]—we removed 20% of the sites that contributed most to compositional heterogeneity[28]. A phylogenetic tree was inferred from the compositionally trimmed supermatrix alignment with IQTREE v1.6.9 (-m LG+C60+F+G, selected by ModelFinder, -bb 1000 -nm 250; Supplementary Fig. 1)[73].

**Binning of metagenomic contigs.** For each metagenome assembly, contigs larger than 2 kb were grouped into bins on the basis of differential coverage across samples, tetranucleotide frequency profiles, GC composition and read-pair linkage as previously described[25]. The contigs were cut every 10 kb, unless the remaining fragment was shorter than 20 kb. Then the preprocessed reads of a set of sequencing runs (125_MIX_0.22-3: all sequencing runs listed in Supplementary Data 2; 067_SRF_0.22-0.45: sequencing runs ERR598994, -599144, -594313, -594325, -594395 and -594404; Supplementary Data 2) were mapped onto the fragmented contigs with KALLISTO v0.42.5[74], yielding differential coverage profiles per fragmented contig. This was then used together with tetranucleotide frequency information by CONCOCT v0.4.0[75] to group the fragmented contigs into bins. Bins containing the Rickettsiales ribocontigs (Fig. 1) were then assessed and cleaned with MMGENOME (accessed June 2016)[76] using differential coverage, GC composition, read-pair linkage and presence of 139 genes well-conserved across Bacteria. Finally, the fragmented contigs of the cleaned bins were replaced by their corresponding full-length contigs. In case not all fragmented contigs from a corresponding full-length contig were present in a cleaned bin, the full-length contig would only be included in the final bin if the majority of the fragmented contigs were present. This yielded three draft genome bins: 'BIN125', 'BIN67-1' and 'BIN67-3'.

We aimed to improve the quality of the draft bins by recruiting reads from all *Tara* Oceans metagenomes that had sequence coverage for the (BIN125: ERR594323, -599156, -594338, -594339, -59434; BIN67-1 and BIN67-3: ERR594395, -594404, -598994, -599144, -594313, -594325) and performing a second round of assembly and binning. This was completed as follows. Preprocessed reads putatively derived from the genomes of interest were recruited from the selected metagenomes by classifying them with Bowtie2 v2.3[77] and CLARK-S v1.2.3[78] using a set of reference Rickettsiales genomes. The recruited reads were combined in two separate pools, one for BIN125 samples and another for BIN67-1/BIN67-3 samples. Each pool was assembled separately with SPAdes (–careful)[79]. The final BIN125 bin was obtained by removing all contigs <3,300 bp. The final BIN67-1 and BIN67-3 bins were obtained by separating the contigs ≥1,500 bp into two groups with CONCOCT (–clusters 4)[75].

**Completeness and redundancy estimates.** We used the miComplete tool v.1.1.1[80] to estimate the completeness and redundancy of the MAGs as well as the reference genomes using a set of bacterial marker genes.

**Annotation.** All bins were annotated with prokka v1.12[81], which was altered to allow for partial gene predictions on contig-edges (GitHub pull request no. 219), with the options–compliant,–partialgenes–cdsrnaolap and–evalue 1e-10, and with barrnap (https://github.com/tseemann/barrnap) as the rRNA predictor. We used eggNOGmapper v1.0.3[82] to get annotations from the EggNOG database 4.5.1[83] (from which we gathered the alphaNOGs). We assigned KEGG[84] orthology (KO) and enzyme commision (EC) numbers using GhostKOALA v2.2[85]. Additionally, we annotated the proteins using the CarbohydrateActive enZYmes Database[86] (CAZY, using HMMER v3.3[87]), the Transporter Classification Database[88] (TCDB, using BlastP 2.8.1+[89]) and used InterProScan v5.42-78.0[90] to annotate the proteins with PFAM[91], TIGRFAM[92] and IPR domains. For detailed annotation of secretion systems and filamentous structures, we screened proteomes using MacSyFinder[93,94] v2.0rc1 with the 'TXSScan'[93,94] and 'TFF-SF'[95] HMM models with '—db_type unordered'. Finally, we used DIAMOND v2.0.6.144[96] to perform similarity searches of the proteins against the non-redundant protein sequence database and recorded the taxonomic annotation of the last common ancestor of all hits within 2% of the best score. Further searches of PFAM[91] domain profiles

were performed using HMMER (-E 1e-05)[87] for eukaryotic domains commonly found in Rickettsiales T4SS effectors (Ankyrin repeats: CL0465; leucin rich repeats: CL0022; Tetratricopeptide repeats: CL0020, Pentapeptide: CL0505)[11,18] as well as in *Xanthomonas* bacterial-killing T4SS effectors (PGB domains: CL0244; Peptidase_M23: PF01551)[45,46]. All annotations are available from the Figshare repository (https://doi.org/10.6084/m9.figshare.c.5494977, see Data availability). We finally searched a selection of genomes for candidate sequences of virB7 homologues using tblastN[89] and references ('rickettsiales+AND+virB7') from Uniprot[97].

**Alphaproteobacteria and mitochondria species tree.** A phylogenomics dataset was constructed by updating the '24 alphamitoCOGs with more diverse mitochondria' dataset from Martijn et al.[25] (henceforth 'alphamito24') with the 3 Rickettsiales MAGs reconstructed in this study, eight Rickettsiales MAGs identified from public MAG datasets[21–24], 'Ca. Deianiraea vastatrix'[17], endosymbiont of *Peranema trichophorum*, and endosymbiont of *Stachyamoeba lipophora*[29] and 'Ca. Phycorickettsia trachydisci'[98]. Orthologues were identified by PSI-BLAST (*E*-value cut-off: $1 \times 10^{-6}$)[89] using the gene alignments of the alphamito24 dataset as queries. Non-orthologues were detected and removed via single-gene tree inspections (alignment: MAFFT L-INS-i v7.471[99]; alignment trimming: trimAl v1.4.rev15 -gappyout[100]; phylogenetic inference: IQTREE v1.6.12 -fast -m LG+F+G[73]). A supermatrix alignment was prepared from the updated orthologous groups by re-aligning with MAFFT L-INS-i[99] and trimming the alignments with BMGE v1.12 -m BLOSUM30[101] before concatenation. The alignment was finalized by removing the 20% most compositionally heterogeneous sites with the $\chi^2$-trimmer[25,28]. A phylogenetic tree was inferred under the posterior mean site frequency (PMSF) approximation[102] of the LG+C60+F+$\Gamma$4 model (selected by ModelFinder[103]; guidetree: LG+G+F, 100 non-parametric bootstraps).

**Genome-based obligate intracellular lifestyle prediction.** We uploaded all Rickettsiales MAG and reference genome proteome files to the PhenDB[52] web server (http://phendb.org/) and used default parameters for prediction of obligate intracellular lifestyles.

**Rickettsiales species tree.** The '129-panorthologues' phylogenomics dataset of Martijn et al.[26] was updated with the same MAGs and genomes that were used to update the alphamito24 dataset, as well as the recently sequenced Rickettsiales 'UBA6177'[23], 'Ca. Xenolissoclinum pacificiensis'[104], 'Ca. Fokinia solitaria'[105], 'Ca. Neoehrlichia lotoris' (ASM96479v1), *Neorickettsia helminthoeca* (ASM63298v1), 'Ca. Jidaibacter acanthamoeba'[27], endosymbiont of *Acanthamoeba* UWC8[72], *Occidentia massiliensis* Os18[106], Rickettsiales bacterium Ac37b and alphaproteobacterial outgroups *Caulobacter crescentus* CB15, 'Ca. Puniceispirillum marinum' IMCC1322[107], *Azospirillum brasiliense* Sp245 (now *Azospirillum baldaniorum*)[108], MarineAlpha3 Bin5, MarineAlpha3 Bin2, MarineAlpha12 Bin1, MarineAlpha11 Bin1, MarineAlpha9 Bin6 and MarineAlpha10 Bin2[25]. Orthologues were identified through PSI-BLAST v2.8.1+ (*E*-value cut-off: $1 \times 10^{-6}$) searches using the 129 gene alignments as a query, and non-orthologues were detected and removed via single-gene tree inspections as described above. A discordance filter[109] was applied to remove the most discordant genes as follows: (i) single-gene alignments were prepared with MAFFT E-INS-i[99] and trimAl v1.4.rev15 -gappyout[100], (ii) single-gene trees were inferred with IQTREE v.1.6.9 (with -bnni)[73,110], (iii) bipartition count profiles were constructed from the bootstraps (tre_make_splits.pl) and compared between all possible gene pairs to calculate discordance scores (tre_discordance_two.pl). The top 13 most discordant genes were removed from the dataset (Supplementary Fig. 2). After preliminary phylogenomics analyses with the remaining 116 genes, we decided to omit extremely long-branching taxa 'Ca. Xenolissoclinum pacificiensis' and 'Ca. Fokinia solitaria' and phylogenetically unstable taxa (endosymbiont of *Stachyamoeba lipophora* and UBA6177) from downstream phylogenomics analyses (Supplementary Data 1). The resultant 116 orthologous groups were first aligned by applying PREQUAL v1.01[111] (masking of putative non-homologous sites), MAFFT E-INS-i (multiple sequence alignment) and Divvier -partial[112] (alignment 'divvying') and then concatenated into an 'untreated' supermatrix alignment. Another 'no-Deianiraea' supermatrix alignment was prepared in an identical manner but with the long-branched 'Ca. D. vastatrix' omitted. From the untreated supermatrix, an iterative $\chi^2$-trimmed alignment (47 rounds of removing the top 1% most heterogeneous sites as determined by $\chi^2$-score[28]) was prepared. The iterative $\chi^2$-trimmer was found to be more efficient at reducing compositional heterogeneity compared with the standard $\chi^2$-trim method[25,28,113]. These were used for phylogenetic reconstruction under the CAT+GTR+$\Gamma$4 (untreated) and CAT+LG+$\Gamma$4 (iterative $\chi^2$-trimmed) models with PhyloBayes MPI v1.8[114]. Four independent Markov chain Monte Carlo (MCMC) chains were run until convergence was reached (maxdiff <0.3) or a sufficient effective sample size was reached (effsize >300), while using a burn-in of at least 5,000 generations. Posterior predictive checks were performed to check to what degree the inferred phylogenetic models captured the across-taxa compositional heterogeneity and site-specific pattern diversity present in the alignments. Parameter configurations were sampled every 50 generations after the burn-in. Maximum likelihood phylogenetic reconstructions were done under the PMSF approximation (with 100 non-parametric bootstraps; guidetree under LG+G+F) of the LG+C60+F+$\Gamma$4 model (selected by ModelFinder) for both supermatrix alignments with IQTREE v1.6.5.

**Gene family trees.** We used the annotation of AlphaNOGs from EggNOG 4.5.1[83] to assign proteins from the MAGs and reference genomes into clusters. All proteins without AlphaNOG annotation were subjected to all-versus-all BLASTP analysis and subsequent de novo clustering with Silix (overlap 90, identity 60)[115], resulting in a total of 34,361 clusters. We computed alignments for all protein clusters with at least 4 members (4,240), using PREQUAL[111] to prefilter unaligned sequences, MAFFT E-INS-i[99] to perform alignments and Divvier v1.01 (-divvygap)[112] to filter alignments. Single-gene trees were inferred for the 4,240 alignments with IQTREE v1.6.9[73] with 1,000 ultrafast bootstraps (-bb 1000 -wbtl)[110]. A model test[103] was performed (-m TESTNEW -mset LG -madd LG+C10,..,LG+C60) for each tree inference. For clusters with only 2 or 3 members, we created 'dummy' bootstraps that represent the only possible topology for an unrooted tree.

**Gene tree-species tree reconciliation.** The 4,240 single-gene trees were reconciled with the species tree derived from the iterative $\chi^2$-trimmed alignment with CAT+LG+$\Gamma$4 using the ALEml_undated algorithm of the ALE suite v0.4[54]. ALE infers gene duplications, losses, transfers, originations and ancestral gene contents along a species tree[53,54]. It also takes into account the completeness of the extant genomes. Per node, a gene family was considered present if 'copies' are ≥0.3 in the ALE output. In addition, gene family evolution events (transfers, originations and losses) were counted similarly, with a threshold of 0.3. Copy numbers and evolutionary events per node in the species tree and gene family are reported by ALE as relative frequencies. These relative frequency values express the support of whether and how many times an evolutionary event occurred in a node (or a gene family was present), while incorporating all the uncertainty of the reconstructed gene tree sample. Therefore, standard statistical thresholds do not apply. We selected 0.3 as a minimal relative frequency due to its high signal to noise ratio (Supplementary Fig. 14). This threshold ensures that events that were reconstructed with a low frequency are still detected, since the signal in single-gene trees can sometimes be very low and events could be overlooked at more stringent thresholds[113,116]. Singleton clusters were counted as originations for the corresponding species.

**Environmental diversity.** The 16S rRNA gene sequences of the MAGs (Gamibacteraceae and Mitibacteraceae, no 16S sequences were found for Athabascaceae) were used to search the NCBI nt database with BLASTN v2.8.1+ (*E*-value <0.05, length >700 bp). An alignment was prepared for Rickettsiales and an alphaproteobacterial outgroup using MAFFT E-INS-i[99], and trimmed with trimAl v1.4.rev15 (-automated1)[100]. A phylogenetic tree was inferred with IQTREE[73] with 100 non-parametric bootstraps under the GTR+F+R8 model (selected by IQTREE's model test[103]).

**T4SS subunit gene trees.** Trees for the T4SS subunits virB1-6,8-11,D4 were prepared using the corresponding eggNOG v.5.0[117] clusters for all Bacteria. First, we removed all Rickettsiales sequences from these clusters to avoid self-alignment. We then used the sequences from our taxon selection as queries to search the COGs using DIAMOND (–top 50–ultra-sensitive)[96] and clustered the obtained sequences using cd-hit[118] at 80% identity. We then aligned these reference sequences together with the sequences from our taxon sampling using MAFFT-E-INS-i[99] and applied a light gap trimming (trimAl v1.4.rev15 -gt 0.01[100]). Finally, trees were inferred in IQTREE[119] with automatic model selection[103] and support values were estimated by 1,000 ultrafast bootstraps (see Data availability).

**Ancestral gene content reconstruction.** We reconstructed ancestral gene family repertoires with the results from ALE by selecting all gene families predicted to be present at a given node with a frequency ≥0.3. Consensus annotations for gene families were inferred on the basis of the abovementioned gene annotations. Metabolic pathways or enzyme complexes were inferred as present if at least half of the necessary genes were present. We assessed the metabolic capabilities of ancestral genomes using the KEGG[84] Module tool or MetaCyc pathways[120].

**Reporting summary.** Further information on research design is available in the Nature Research Reporting Summary linked to this article.

## Data availability

In addition to data available in the supplementary materials, files containing sequence datasets, alignments and phylogenetic trees in Newick format are archived at the digital repository Figshare: https://doi.org/10.6084/m9.figshare.c.5494977. MAGs generated in this study are linked to BioProject PRJNA746308. Accessions for genomes analysed in this study can be found in Supplementary Data 3. Publicly available datasets include eggNOG v4.5.1 (eggnog45.embl.de/), KEGG (kegg.jp), CAZY (cazy.org), TCDB (tcdb.org), PFAM (pfam.xfam.org), TIGRFAM and InterPro (ebi.ac.uk/interpro/), NCBI nucleotide (ncbi.nlm.nih.gov/nucleotide/) and MetaCyc v.26.0 (biocyc.org/META/).

## Code availability

Custom scripts used are available on GitHub (https://github.com/maxemil/rickettsiales-evolution, https://github.com/maxemil/ALE-pipeline and https://github.com/novigit/broCode).

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

## Acknowledgements

We thank those involved in the generation of metagenomic datasets analysed in the present work and in making these publicly available to the scientific community;

the Uppsala Multidisciplinary Center for Advanced Computational Science (UPPMAX) at Uppsala University and the Swedish National Infrastructure for Computing (SNIC) at the PDC Center for High-Performance Computing for providing computational resources. This work was funded by European Research Council Consolidator (817834), Dutch Research Council (VI.C.192.016), Swedish Research Council (2015-04959), Volkswagen Foundation (96725) and European Union (H2020-MSCA-ITN-2015-675752) grants to T.J.G.E., and by Swedish Research Council (2018-06727) grant to J.M. We acknowledge J. E. Dharamshi for the original idea of the iterative $\chi^2$-trimming as well as for discussions regarding ALE.

## Author contributions

T.J.G.E. conceived and supervised the study. J.M., J.V. and M.E.S. screened, assembled and binned metagenomic datasets. J.M. and M.E.S. performed phylogenomic analyses and ancestral genome reconstruction analyses. M.E.S., J.M. and S.K. analysed ancestral genome content. M.E.S., J.M., S.K. and T.J.G.E. interpreted the obtained results and wrote the first manuscript draft. All authors edited and approved the final version of the manuscript.

## Funding

## Competing interests

The authors declare no competing interests.

## Additional information

**Extended data** is available for this paper at https://doi.org/10.1038/s41564-022-01169-x.

**Correspondence and requests for materials** should be addressed to Thijs J. G. Ettema.

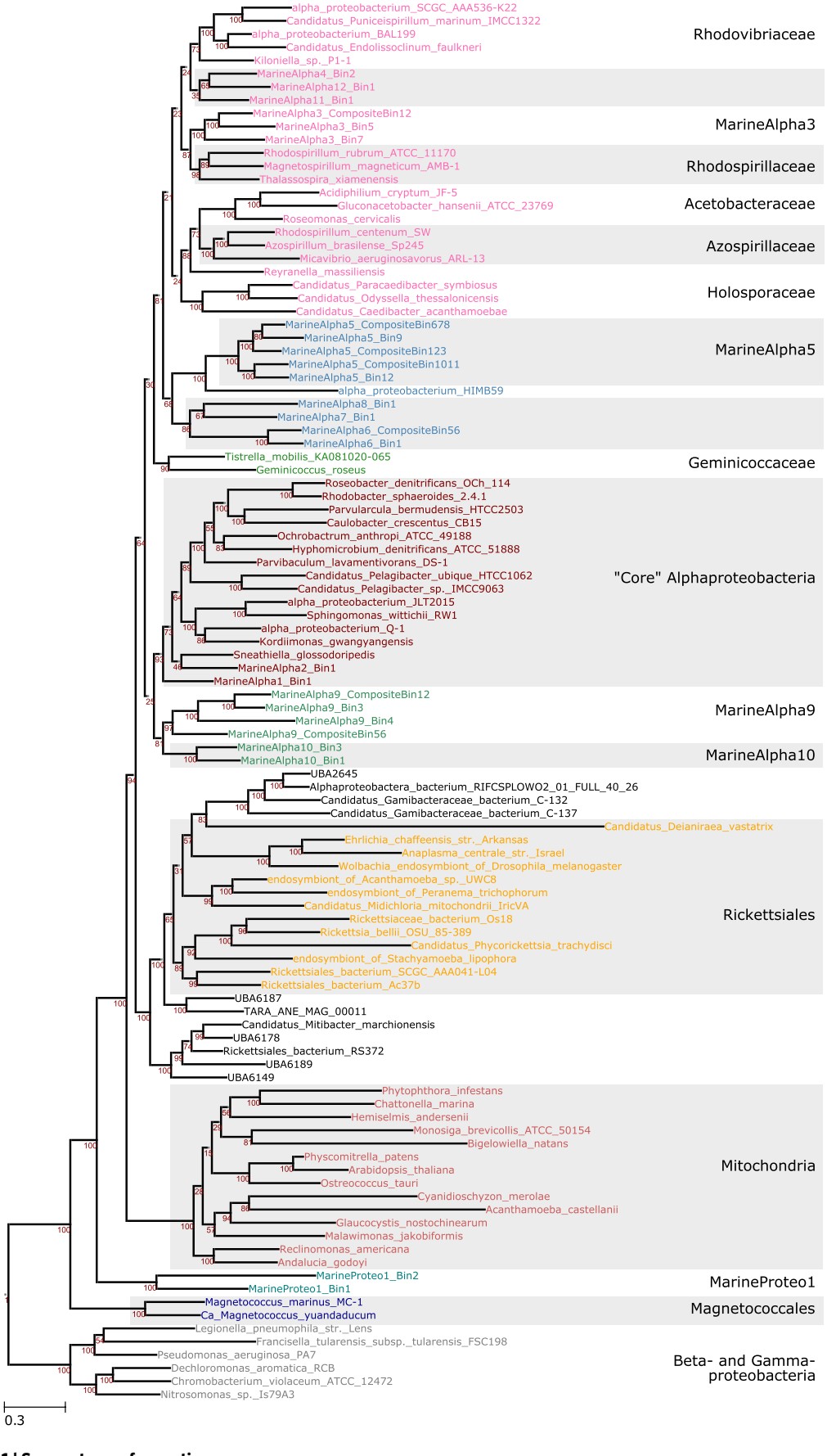

**Extended Data Fig. 1 | See next page for caption.**

**Extended Data Fig. 1 | Maximum likelihood phylogeny of Alphaproteobacteria and mitochondria.** Based on the $\chi^2$-trimmed (20% most heterogeneous sites removed) concatenated alignment of 24 highly conserved mitochondrially encoded proteins. ML tree was inferred under the PMSF approximation of LG+C60+F+$\Gamma$4 with 100 non-parametric bootstraps as implemented by IQ-TREE. Distinct alphaproteobacterial clades are given their own unique color, including the Rickettsiales (orange) and mitochondria (salmon). Rickettsiales MAGs identified and/or reconstructed in this study are given in black.

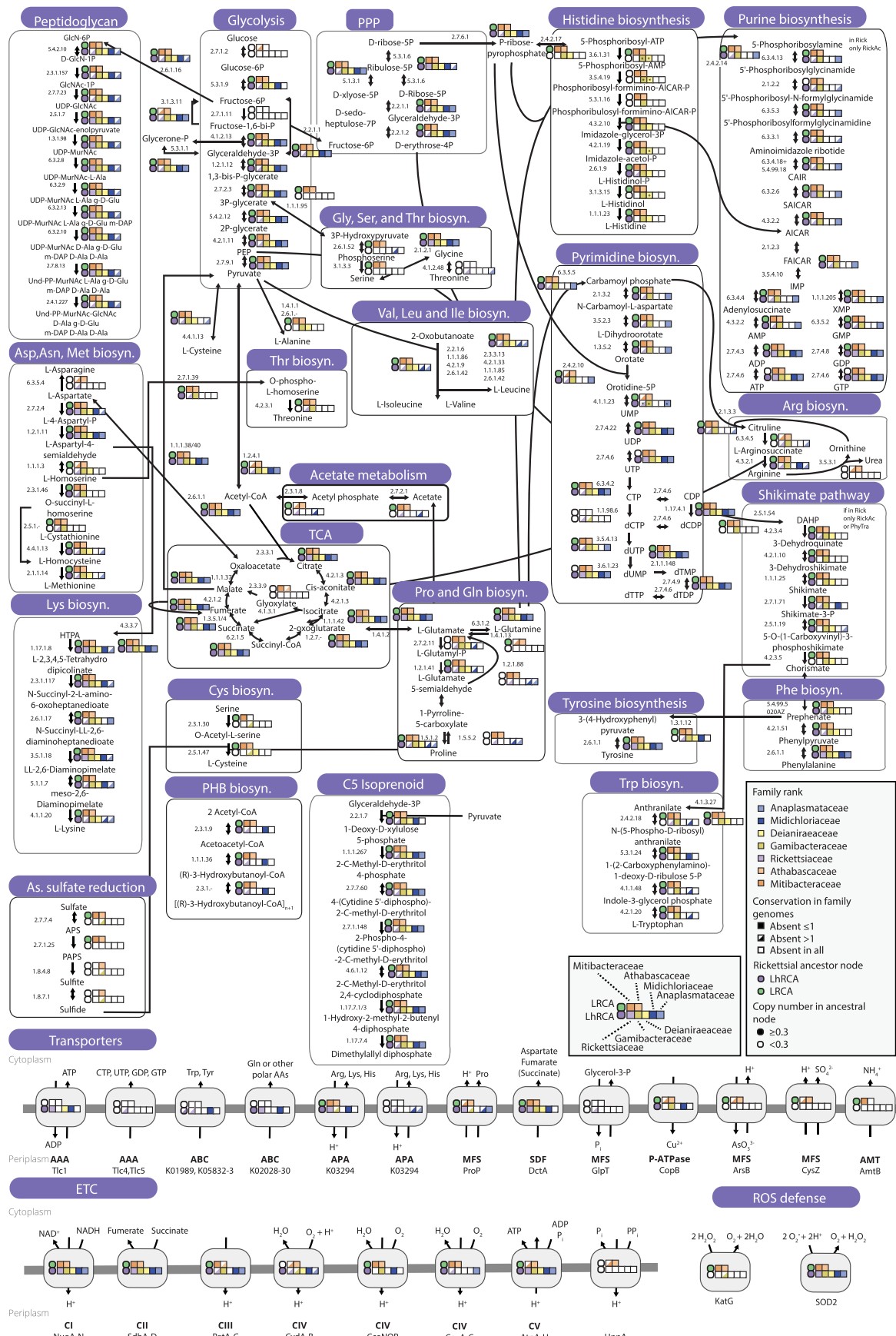

**Extended Data Fig. 2 | See next page for caption.**

**Extended Data Fig. 2 | Overview of conservation of central metabolism, transporters, and electron transport chain (ETC) in Rickettsiales families and key ancestors.** All enzymatic steps are depicted with enzyme commision (EC) numbers. The corresponding enzyme name and distribution in Rickettsiales can be found in Supplementary Data 4. Arrows between molecules and arrow heads indicate catalyzed enzymatic reactions and directionality, respectively. Presence/absence is indicated for extant genomes of Anaplasmataceae (light blue), Midichloriaceae (dark blue), Deianiraeaceae (light yellow), Gamibacteraceae (dark yellow), Rickettsiaceae (light violet), Athabascaceae (light orange), Mitibacteraceae (dark orange) as well as the reconstructed genomes of the ancestor of all obligate host-associated (classical) Rickettsiales (LhRCA, dark violet) and the ancestor of all Rickettsiales including Mitibacteraceae and Athabascaceae (LRCA, green). For the seven families, full squares represent absence in at most one member, half squares absence in more then one member but presence in some, and empty squares absence in all members. For reconstructed ancestral genomes, genes are inferred as present if the ALE program inferred an ancestral relative frequency of at least 0.3.

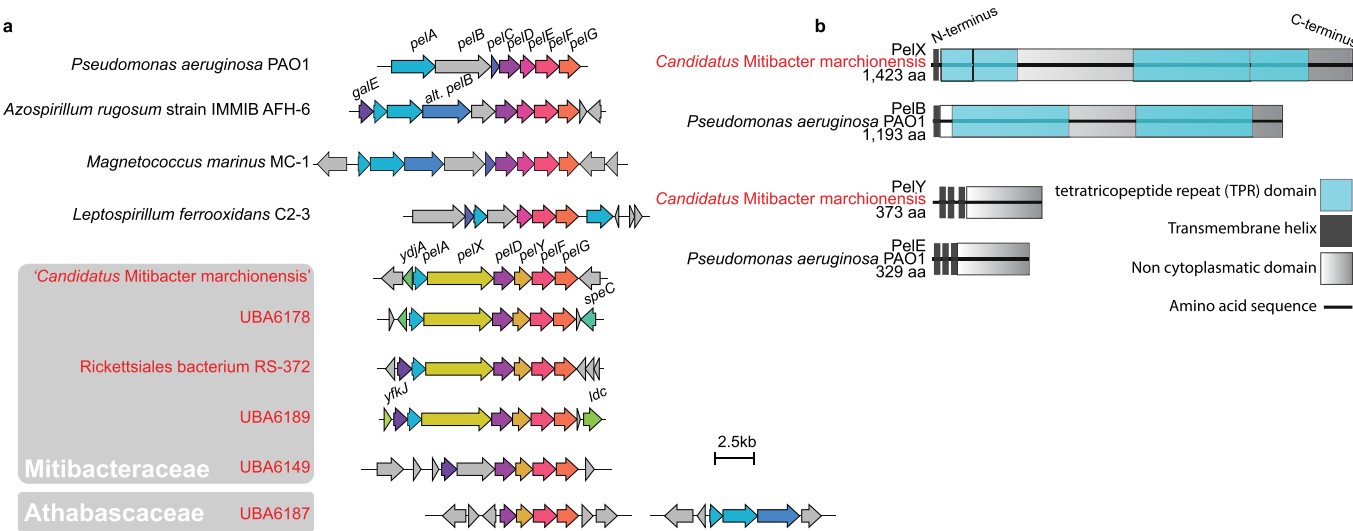

**Extended Data Fig. 3 | Gene synteny of and domain architecture of derived pel exopolysaccharide gene cluster in free-living Rickettsiales.** (a) Gene synteny plot of *pel*ABCDEFG gene cluster in the model organism *Pseudomonas aeruginosa* PAO, and related clusters in selected free-living bacteria, and free-living Rickettsiales. Arrows indicate the genomic orientation and size of genes. Scale bar indicates a length of 2.5 kb. (b) Domain architecture of likely functional equivalents of the *P. aeruginosa* PelB and PelE proteins in the Rickettsiales bacterium *Candidatus* Mitibacter marchionensis.

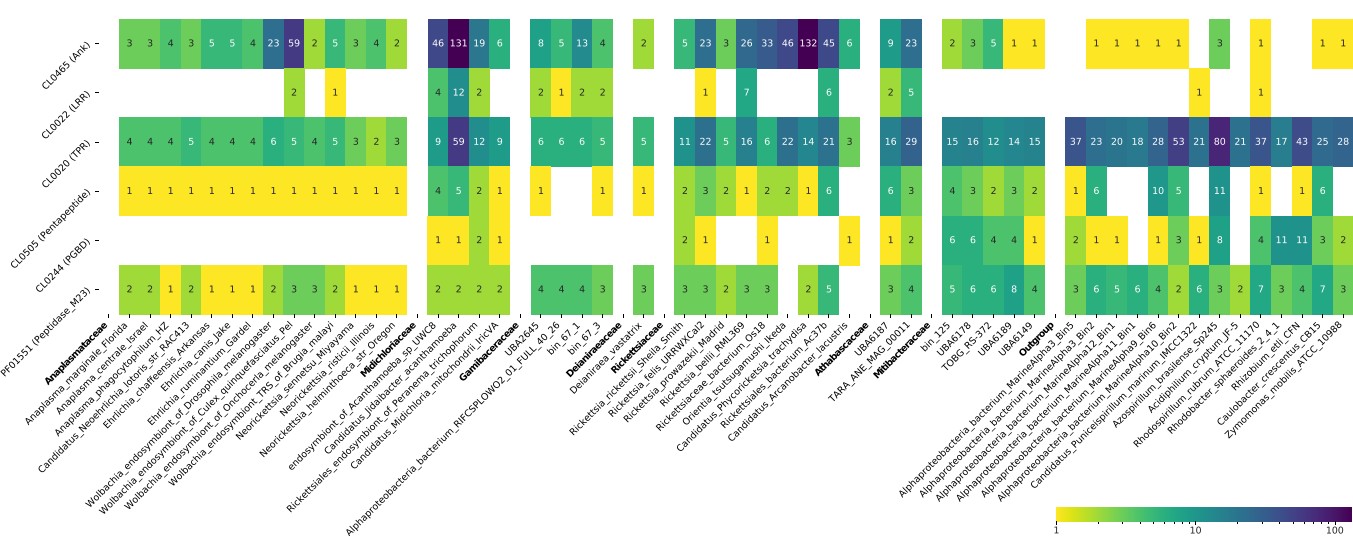

**Extended Data Fig. 4 | Frequency of proteins containing selected PFAM domain families.** Domains typical of Rickettsiales effectors (Ank, LRR, TPR) and domains often found in *Xanthomonas* bacterial-killing effectors (Peptidoglycan binding domain PGBD, Peptidase M23) in the selected taxa used for phylogenomic and ancestral reconstruction. Domains were identified using HMMER hmmsearch (-E 1e-05) of the corresponding Pfam domain profiles.

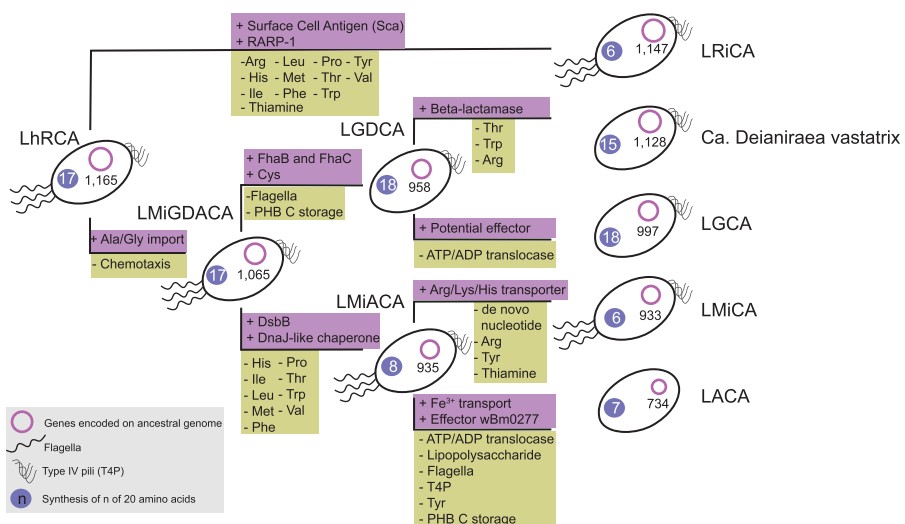

**Extended Data Fig. 5 | Key gains, losses and characteristics of host-associated Rickettsiales ancestors.** Schematic phylogenetic tree depicting the relationships of the last common ancestor of host-associated Rickettsiales (LhRCA) and Rickettsiales families ancestors. Last common ancestor of Anaplasmataceae, Midichloriaceae, Deianiraea and Gamibacteraceae: LMiGDACA; Last common ancestor of Anaplasmataceae and Midichloriaceae: LMiACA; Last common ancestor of Anaplasmataceae: LACA, Midichloriaceae: LMiCA, Gamibacteraceae: LGCA; Rickettsiaceae: LRiCA. Gains (violet) and losses (green) of genes encoding for key characteristics are written above and below the corresponding branch, respectively. The ancestors are depicted as cells with inferred features such as the presence of a flagellum, Type 4 pili, the capacity to synthesize amino acids and the number of inferred ancestral genes. Amino acid biosynthesis pathways are represented by the three-letter code of the produced amino acid. Proteins related to disulfide bond formation (DsbB); filamentous hemagglutinin production (FhaB), secretion and activation (FhaC); rickettsial ankyrin repeat protein (RARP-1); surface cell antigen (Sca); polyhydroxybutyrate (PHB) synthesis.

# Reporting Summary

## Statistics

For all statistical analyses, confirm that the following items are present in the figure legend, table legend, main text, or Methods section.

| n/a | Confirmed | |
|---|---|---|
| ☐ | ☒ | The exact sample size (*n*) for each experimental group/condition, given as a discrete number and unit of measurement |
| ☒ | ☐ | A statement on whether measurements were taken from distinct samples or whether the same sample was measured repeatedly |
| ☒ | ☐ | The statistical test(s) used AND whether they are one- or two-sided<br>*Only common tests should be described solely by name; describe more complex techniques in the Methods section.* |
| ☒ | ☐ | A description of all covariates tested |
| ☒ | ☐ | A description of any assumptions or corrections, such as tests of normality and adjustment for multiple comparisons |
| ☒ | ☐ | A full description of the statistical parameters including central tendency (e.g. means) or other basic estimates (e.g. regression coefficient) AND variation (e.g. standard deviation) or associated estimates of uncertainty (e.g. confidence intervals) |
| ☐ | ☒ | For null hypothesis testing, the test statistic (e.g. *F*, *t*, *r*) with confidence intervals, effect sizes, degrees of freedom and *P* value noted<br>*Give P values as exact values whenever suitable.* |
| ☒ | ☐ | For Bayesian analysis, information on the choice of priors and Markov chain Monte Carlo settings |
| ☒ | ☐ | For hierarchical and complex designs, identification of the appropriate level for tests and full reporting of outcomes |
| ☒ | ☐ | Estimates of effect sizes (e.g. Cohen's *d*, Pearson's *r*), indicating how they were calculated |

*Our web collection on statistics for biologists contains articles on many of the points above.*

## Software and code

Policy information about availability of computer code

| Data collection | No software was used for data collection |
|---|---|
| Data analysis | SEQPREP v1.3.2, TRIMMOMATIC v0.35, FASTQC v0.11.4, metaSPAdes (SPAdes 3.7.0), Prodigal v2.60, MAFFT v7.471, trimAl v1.4.rev15, IQTREE v1.6.9, v16.12, v2.0, KALLISTO v0.42.5, CONCOCT v0.4.0, MMGENOME (as on GitHub june 2016), Bowtie2 v2.3, CLARK-S v1.2.3, miComplete v.1.1.1, prokka v1.12,<br>eggNOG-mapper v1.0.3, GhostKOALA (KEGG tools) v2.2, HMMER v3.3, InterProScan v5.42-78.0, MacSyFinder v2.0rc1, DIAMOND v2.0.6.144, PSI-BLAST<br>v2.8.1+, BMGE v1.12, PREQUAL v1.01, Divvier v1.01, PhyloBayes MPI v1.8, ALE v0.4, BLASTN v2.8.1+, BLASTP v2.8.1+<br><br>Custom code:<br>https://github.com/maxemil/rickettsiales-evolution,<br>https://github.com/maxemil/ALE-pipeline,<br>https://github.com/novigit/broCode |

For manuscripts utilizing custom algorithms or software that are central to the research but not yet described in published literature, software must be made available to editors and reviewers. We strongly encourage code deposition in a community repository (e.g. GitHub). See the Nature Portfolio guidelines for submitting code & software for further information.

## Data

Policy information about availability of data

All manuscripts must include a data availability statement. This statement should provide the following information, where applicable:

- Accession codes, unique identifiers, or web links for publicly available datasets
- A description of any restrictions on data availability
- For clinical datasets or third party data, please ensure that the statement adheres to our policy

In addition to data available in the supplementary materials, files containing sequence datasets, alignments, and phylogenetic trees in Newick format are archived at the digital repository Figshare: 10.6084/m9.figshare.c.5494977. MAGs generated in this study are linked to BioProject PRJNA746308. Accessions for genomes analyzed in this study can be found in Supplementary Data 3. Publicly available datasets include eggNOG v4.5.1 (eggnog45.embl.de/), KEGG (kegg.jp), CAZY (cazy.org), TCDB (tcdb.org), PFAM (pfam.xfam.org), TIGRFAM and InterPro (ebi.ac.uk/interpro/), NCBI nucleotide (ncbi.nlm.nih.gov/nucleotide/), MetaCyc v.26.0 (biocyc.org/META/)

# Field-specific reporting

Please select the one below that is the best fit for your research. If you are not sure, read the appropriate sections before making your selection.

☐ Life sciences  ☐ Behavioural & social sciences  ☒ Ecological, evolutionary & environmental sciences

For a reference copy of the document with all sections, see nature.com/documents/nr-reporting-summary-flat.pdf

# Ecological, evolutionary & environmental sciences study design

All studies must disclose on these points even when the disclosure is negative.

| | |
|---|---|
| Study description | Reconstruction of novel alphaproteobacterial genomes (MAGs). Phylogenomic analyses of Rickettsiales and related MAGs. Gene-tree species tree reconciliation and ancestral reconstruction of the last common ancestor of Rickettsiales. |
| Research sample | Available metagenomic datasets from the Tara Oceans consortium, published MAGs and reference genomes of Rickettsiales and alphaproteobacteria |
| Sampling strategy | We selected particular metagenomic datasets of the Tara Oceans consortium and the other MAGs based on a phylogenetic screen of contigs containing ribosomal protein genes. Those datasets and MAGs that contained contigs related to Rickettsiales were selected |
| Data collection | N/A because the primary data collection was done by other parties (i.e. the Tara Oceans expedition and other research groups that collected the raw sequence data underlying the MAGs we selected) |
| Timing and spatial scale | N/A because we did not do the primary data collection |
| Data exclusions | Certain Rickettsiales taxa were excluded from phylogenetic analyses as they were found to have extremely long branches and their inclusion would lead to untrustworthy results due to long branch attraction artefacts. Criteria for excluding were not pre-established |
| Reproducibility | All results of this study can be reproduced given the same original source data and the methods provided in this manuscript |
| Randomization | N/A because randomization was not required for the purposes of this study |
| Blinding | N/A because blinding was not required for the purposes of this study |

Did the study involve field work?  ☐ Yes  ☒ No

# Reporting for specific materials, systems and methods

We require information from authors about some types of materials, experimental systems and methods used in many studies. Here, indicate whether each material, system or method listed is relevant to your study. If you are not sure if a list item applies to your research, read the appropriate section before selecting a response.

## Materials & experimental systems

| n/a | Involved in the study |
|---|---|
| ☒ | Antibodies |
| ☒ | Eukaryotic cell lines |
| ☒ | Palaeontology and archaeology |
| ☒ | Animals and other organisms |
| ☒ | Human research participants |
| ☒ | Clinical data |
| ☒ | Dual use research of concern |

## Methods

| n/a | Involved in the study |
|---|---|
| ☒ | ChIP-seq |
| ☒ | Flow cytometry |
| ☒ | MRI-based neuroimaging |

