## [Peer Review File · Nature Microbiology]

Peer Review Information

Journal: Nature Microbiology

Manuscript Title: The evolutionary origin of host association in an ancient bacterial clade

Corresponding author name(s): Thijs Ettema

Reviewer Comments & Decisions:Decision Letter, initial version:

Dear Professor Ettema,

Thank you for your patience while your manuscript "The evolutionary origin of host association in an ancient bacterial clade" was under peer-review at Nature Microbiology. It has now been seen by 3 referees, whose expertise and comments you will find at the of this email. You will see from their comments below that while they find your work of interest, some important points are raised. We are very interested in the possibility of publishing your study in Nature Microbiology, but would like to consider your response to these concerns in the form of a revised manuscript before we make a final decision on publication.

In particular, you will see that the reviewers are positive about your work and they mostly raise minor comments and clarifications that need to be addressed. Most of the remaining issues should be straightforward to address.

If you have not done so already please begin to revise your manuscript so that it conforms to our Article format instructions at <http://www.nature.com/nmicrobiol/info/final-submission/>

The usual length limit for a Nature Microbiology Article is six display items (figures or tables) and 3,000 words. We have some flexibility, and can allow a revised manuscript at 3,500 words, but please consider this a firm upper limit. There is a trade-off of ~250 words per display item, so if you need more space, you could move a Figure or Table to Supplementary Information.

Some reduction could be achieved by focusing any introductory material and moving it to the start of your opening 'bold' paragraph, whose function is to outline the background to your work, describe in a sentence your new observations, and explain your main conclusions. The discussion should also be limited. Methods should be described in a separate section following the discussion, we do not place a word limit on Methods.

Nature Microbiology titles should give a sense of the main new findings of a manuscript, and should not contain punctuation. Please keep in mind that we strongly discourage active verbs in titles, and that they should ideally fit within 90 characters each (including spaces).

Please include a data availability statement as a separate section after Methods but before

2references, under the heading "Data Availability". This section should inform readers about the availability of the data used to support the conclusions of your study. This information includes accession codes to public repositories (data banks for protein, DNA or RNA sequences, microarray, proteomics data etc...), references to source data published alongside the paper, unique identifiers such as URLs to data repository entries, or data set DOIs, and any other statement about data availability. At a minimum, you should include the following statement: "The data that support the findings of this study are available from the corresponding author upon request", mentioning any restrictions on availability. If DOIs are provided, we also strongly encourage including these in the Reference list (authors, title, publisher (repository name), identifier, year). For more guidance on how to write this section please see:

<http://www.nature.com/authors/policies/data/data-availability-statements-data-citations.pdf>

To improve the accessibility of your paper to readers from other research areas, please pay particular attention to the wording of the paper's opening bold paragraph, which serves both as an introduction and as a brief, non-technical summary in about 150 words. If, however, you require one or two extra sentences to explain your work clearly, please include them even if the paragraph is over-length as a result. The opening paragraph should not contain references. Because scientists from other sub-disciplines will be interested in your results and their implications, it is important to explain essential but specialised terms concisely. We suggest you show your summary paragraph to colleagues in other fields to uncover any problematic concepts.

If your paper is accepted for publication, we will edit your display items electronically so they conform to our house style and will reproduce clearly in print. If necessary, we will re-size figures to fit single or double column width. If your figures contain several parts, the parts should form a neat rectangle when assembled. Choosing the right electronic format at this stage will speed up the processing of your paper and give the best possible results in print. We would like the figures to be supplied as vector files - EPS, PDF, AI or postscript (PS) file formats (not raster or bitmap files), preferably generated with vector-graphics software (Adobe Illustrator for example). Please try to ensure that all figures are non-flattened and fully editable. All images should be at least 300 dpi resolution (when figures are scaled to approximately the size that they are to be printed at) and in RGB colour format. Please do not submit Jpeg or flattened TIFF files. Please see also 'Guidelines for Electronic Submission of Figures' at the end of this letter for further detail.

Figure legends must provide a brief description of the figure and the symbols used, within 350 words, including definitions of any error bars employed in the figures.

Finally, please ensure that you retain unprocessed data and metadata files after publication, ideally archiving data in perpetuity, as these may be requested during the peer review and production

process or after publication if any issues arise.

Please include a statement before the acknowledgements naming the author to whom correspondence and requests for materials should be addressed.

Finally, we require authors to include a statement of their individual contributions to the paper -- such as experimental work, project planning, data analysis, etc. -- immediately after the acknowledgements. The statement should be short, and refer to authors by their initials. For details please see the Authorship section of our joint Editorial policies at http://www.nature.com/authors/editorial_policies/authorship.html

- * include a point-by-point response to any editorial suggestions and to our referees. Please include your response to the editorial suggestions in your cover letter, and please upload your response to the referees as a separate document.

- * ensure it complies with our format requirements for Letters as set out in our guide to authors at www.nature.com/nmicrobiol/info/gta/

- * state in a cover note the length of the text, methods and legends; the number of references; number and estimated final size of figures and tables

- * resubmit electronically if possible using the link below to access your home page:

{redacted}

*This url links to your confidential homepage and associated information about manuscripts you may have submitted or be reviewing for us. If you wish to forward this e-mail to co-authors, please delete this link to your homepage first.

Please ensure that all correspondence is marked with your Nature Microbiology reference number in the subject line.

Nature Microbiology is committed to improving transparency in authorship. As part of our efforts in this direction, we are now requesting that all authors identified as 'corresponding author' on published papers create and link their Open Researcher and Contributor Identifier (ORCID) with their account on the Manuscript Tracking System (MTS), prior to acceptance. This applies to primary research papers only. ORCID helps the scientific community achieve unambiguous attribution of all scholarly contributions. You can create and link your ORCID from the home page of the MTS by clicking on 'Modify my Springer Nature account'. For more information please visit www.springernature.com/orcid.

We hope to receive your revised paper within three weeks. If you cannot send it within this time, please let us know.

Yours sincerely,

{redacted}

Reviewer Expertise:

Referee #1: Rickettsiales/microbial genomics

Referee #2: host-obligate pathogens biology/Rickettsia

Referee #3: host-microbe symbiosis

Reviewers Comments:

Reviewer #1 (Remarks to the Author):

Fantastic job! My comments are mostly minor and meant to provide some resolution where it may be appreciated, and suggestions on clarity. This is a massive effort and will have a strong impact on Rickettsiology. Kudos!

The manuscript titled "The evolutionary origin of host association in an ancient bacterial clade" by Schön and colleagues reports on the discovery of novel lineages of Order Rickettsiales by 1) assembling 11 new genomes from metagenomic datasets obtained from aquatic environments, and 2) conducting robust phylogenomics analyses to define features of the new and existing lineages. Prior to this report, Rickettsiales consisted of three established families (Rickettsiaceae, Anaplasmataceae, Midichloriaceae) and a recently proposed family (Deianiraeaceae) that forms a very long branch in estimated phylogenies. The Rickettsiaceae, Anaplasmataceae, Midichloriaceae comprise obligate intracellular species, while the Deianiraeaceae was predicted to comprise species with a recent transition from facultative to obligate intracellular. Remarkably, the 11 new assembled genomes substantially added to rickettsial diversity such that two basal lineages were revealed (proposed families Mitibacteraceae and Athabascaceae) as well as a lineage most closely related to Deianiraeaceae (proposed family Gamibacteraceae). Detailed phylogenomics analysis are employed to propose a gradual shift from free-living/biofilm-associated lifestyles in the basal Mitibacteraceae and Athabascaceae species. Specific focus on the Rickettsiales vir homolog (Rvh) type IV secretion system (T4SS) resulted in a hypothesis that this complex has been repurposed throughout rickettsial evolution as this lifestyle shift occurred.

Aside from the human health, biocontrol, and ecological importance of Rickettsiales, no discussions on the evolution of eukaryotes are possible without considering this ancient proteobacterial lineage. Hypotheses on the origin of the mitochondria, particularly the closest extant lineage to the mitochondrial precursor, and subject to change as advances are made in phylogenetic methodology but also as novel species are revealed that help fill in the gaps between extinct and extant lineages. Thus, this work by Schön and colleagues is a substantial contribution to evolutionary biology and the origin of Eukaryota. It also has a profound impact on Rickettsiology, providing insight on how the different lineages have diversified and evolved discrete relationships with eukaryotic host cells. These novel assemblies will allow estimation of robust phylogenetic frameworks to decipher the evolution of traits underpinning endosymbiosis, reproductive parasitism, and vertebrate virulence (among others). Furthermore, the methodological approach by Schön and colleagues will illuminate the power

5of metagenomic datasets (if analyzed with care and caution) for illuminating unrealized diversity and advancing classification for virtually any microbial system. Thus, I predict that this work by Schön and colleagues will make a substantial and lasting impact on the fields of evolutionary biology, Rickettsiology, and genomics/phylogenomics.

The manuscript is very well written and presented in just the right format to keep the salient points at the front and important minutia in the supporting materials. Additionally, the authors make their code and scripts available at github. The manuscript is the right length for Nature Microbiology and should have a broad appeal for readers. Below are mostly minor comments (moving right through the manuscript, so supporting information is discussed as it appears throughout).

Line 43-44: "leading" used twice, stylistic suggestion.

Line 54 or 56: Consider citing PMID: 28951473 for metabolomic reconstruction.

Line 66: "In-depth".

Line 70: I realize pathogenicity is important but so are other host-relationships that evolved, like mutualism and reproductive parasitism.

Line 80: Supp. Figure 1 is very difficult to read. I think the font size should be greatly expanded.

Line 83: Supp. Figure 2 is also very difficult to read. Aside from small font size, the color scheme might be a bit too much. Maybe symbols and boxes around clades to help with the taxonomic assignments?

Line 101-114: Regarding the shortened branch for *Ca. Deianiraea vastatrix*, have you considered removing this taxon altogether and estimating phylogenies on the datasets with the various employed approaches? Has there been an evaluation of the *Ca. Deianiraea vastatrix* genome to determine its quality? Is it possible it is a mosaic assembly? Please consider citing PMID: 23475938 since it showed a similar lack of corroboration for the placement of Midichloriaceae relative to Rickettsiaceae and Anaplasmataceae (different datasets yielded different, albeit weakly supported, topologies). Perhaps there is still some large swath of diversity we are missing from this region of the tree?

Line 119: Why would rDNA sequences not be available? This seems strange that so much other DNA would be present but not rDNA sequences. Are they partial? Are there others available from those metagenomes?

Line 125: The branch support in Supp. Figure 8 is very difficult to read. Also, aren't there many 16S rDNA sequences available for Deianiraeaceae to include?

Line 125: This is reminiscent of PMID: 23475938 wherein reads were assembled from the Trichoplax genome assembly into a putative symbiont (RETA).

Figure 2: The green shadings for the Athabascaceae and Mitibacteraceae are hard to distinguish. Also, is it really OK to call Deianiraeaceae extracellular? Could it not be an obligate epicellular species? I feel like some readers may be confused by this. I think the references to (b) and (c) are swapped in the legend. While the figure is very informative and powerful in its presentation, it is difficult to read and will hopefully be enlarged for publication. You could decrease the width of the

histograms (d-f) to increase the font sizes of all the words.

Supplementary Text: this is a fantastic summary of the phylogenetic methodology and very robust. I see no weakness in the methods or rationale for data exclusion, though the χ^2 -trimming approach could be explained a bit more. In regard to the "Putative T4SS effectors in Mitibacteraceae and Athabascaceae", PMID: 32546622, PMID: 29946049 and PMID: 26291822 should be cited for PK3, ANK and ARF-GEF domains (eukaryotic) in Rickettsia T4SS effectors. In regard to Supp. Fig. 11, the sequences (accession numbers) from the HMMER searches would be nice as a supporting file to help researchers focus on possible new T4SS effectors. I am sure the results are repeatable, so even the exact query sequence used to get the results shown in the table would be great. If multiple queries were used per category it would be better to just provide a table. I found it difficult to navigate between Supp. Data 5 and Supp. Data 6, and while it is cool to map the distribution of these domains, providing the Uniprot/NCBI accession numbers for each sequence is more helpful for the community.

Line 156: Extended Data Figure 1: this is a nice depiction of the metabolic reconstruction; however, the colors are very difficult to assign to the taxa. If this has to remain this small, I suggest an example of the circles/squares with dotted lines connecting the taxa to each ball/square. The lines connection various biochemical processes are difficult to interpret in some cases. What is ALE in the legend?

Line 156: Supp. Data 4: this table is nicely made, but can an NCBI/Uniprot accession be taken from the table? Can a researcher find the protein for a specific taxon easily in this dataset? Again, exemplar sequences (IDs) at least would be helpful. What are the IDs in column A? The legend does not provide enough information.

Line 164: Do the proteins unique to Athabascaceae and Mitibacteraceae show a vertical descent pattern from the closest Rickettsiales lineage?

Line 181: Is the flagellar apparatus the same across the Rickettsiales lineages that possess it? Or have the Midichloriaceae and/or Rickettsiaceae versions been acquired via lateral gene transfer?

Figure 3: Can you switch the direction of the half circle diagonal to distinguish between partial gene presence vs. partial taxon presence? Or do both cases occur per lineage such that it is more complicated? I think "CONJ" is misleading as presented. Some genomes just have remnants of ICEs while others have a full conjugation apparatus. Most of the Rickettsiaceae have degraded ICEs. The numbers for amino acids inside the orange circles are difficult to make out. Maybe try a darker orange and white numbers? Similar as above: do the genes encoding the chemotaxis proteins, T4P, tad, and cbb3-like oxidases show vertical descent from a common ancestor? In panel B, why is RvhB7 not shown? These genes are there, they are just hard to find. Sequences are shown in Figure 4 of PMID: 20176788. I'm sure an HMMER search would find them. P-T4SSs need a lipoprotein in the core complex.

Line 185: It is very difficult to read the text in this figure. Please expand so the font size is legible.

Line 189: Replace "could" with "can"; as is, it sounds like these bacteria are extinct.

Line 192: Supp. Figure 10 is completely unreadable. Can you make the taxon names larger to the left and right of the tree? There appear to be some red sequences outside of the major RvhB4

7duplication. What are these? Also, Figure 2 in PMID: 26646013 showed ancient duplication of the rvh T4SS within a larger T4SS phylogeny.

Line 194: So, "RARP" was named for specific Rickettsia proteins that are secreted (shown experimentally) and carry ankyrin repeats (PMID: 22773786). RARP-1 was shown to be secreted during host infection, and secretion in a heterologous system (E. coli) required a SEC signal sequence and the TolC protein of E. coli. So RARP-1 was proposed to be secreted via two-step translocation. RARP-2 was also shown to be secreted during infection and its C-terminal "tail" interacted with RvhD4 in a B2H assay (PMID: 29946049). A pull-down using RvhD4 antibody also captured RARP-2 (among many other proteins, some of which are predicted to be T4SS effectors).

Line 197: A newer study characterized Risk-1, a Rickettsia secreted PI3 kinase PMID: 32546622. This protein was also captured in the pull-down described above.

Line 199: Are the query sequences available (IDs) used in the HMMER searches?

Line 206-216: I agree that the combined observations support Athabascaceae and Mitibacteraceae as facultative or extracellular. Why, then, is the VirB5 protein absent from these T4SSs? Most P-T4SSs elaborate a pilus that needs the VirB5 protein.

Extended Data Figure 2: the numbers inside the blue circles cannot be seen. Try white numbers. Are the various gene gains supported by phylogeny estimation? In other words, do phylogenies indicate LGT? If so, from where?

Line 253: It could also be a conjugation system or involved in DNA uptake. Both the ancestral and modern form. What is fascinating is the oddity of the RvhB4,B8,B9 duplication and expansion of the large RvhB6 proteins. Why are these present across all of the species even when there are now such distinct lifestyles?

Line 266: So, provided that Ca. D. vastatrix is still "obligate" from a metabolic perspective, perhaps the P-T4SS is involved in sucking nutrients out of cells?

Line 338: I am not sure what is meant by the rickettsial OMP. Rickettsia have an expanded autotransporter family called Surface Cell Antigens (Scas); Orientia has counterparts. If this is what is being referred to, the references for these should be cited. Using "OMP" is very misleading.

Figure 4: Some Rickettsia have the ability to convert acetate to acetyl-CoA using AckA and Pta. Not sure if those are the enzymes referred to in the Losses box.

Line 502: Per my notes above for lacking sequence information: if the Figshare repository has this information, then ignore my comments. I only ask that all figures and tables listing presence/absence of genes allow for repeatability.

Line 602: Is it possible to search again for VirB7 and VirB5? I am not sure why this were not included. It would be cool to find VirB5 in the Athabascaceae and Mitibacteraceae.

Materials and Methods. Excellent. The workflow is explained well and should be repeatable if all sequence information is accessible. What a massive effort!

Reviewer #2 (Remarks to the Author):

This is a very exciting new paper presenting eleven new metagenome assembled genomes of organisms within the Rickettsiales, placed in three clades. At least two of these are likely to represent organisms that are free living or host associated (but not obligate intracellular). This work adds to a growing body of research that is expanding our view of the Rickettsiales beyond the well described families of Rickettsiaceae and Anaplasmataceae. The recent publication of *Ca. Deianiraea vastatrix* gave the first indication of some free-living Rickettsiales, and the current manuscript greatly strengthens this finding. This work will have a profound impact on our understanding of Rickettsiales and the adaptation to obligate intracellular replication. I am very enthusiastic about this manuscript and have only minor comments:

1. Line 80-83: the use of mitochondrially encoded marker genes is not clear on a first reading and should be elaborated.
2. Line 197-198: what do they mean by not being able to detect "increased numbers" of putative effector proteins? Increased relative to what? Were Anks/TPRs detected and if so how many?
3. Line 245: the discussion of the expanded gene repertoire in *Orientia* should include mention of the very high number of pseudogenes.
4. Line 269-271: It would help the reader to include again the number of predicted protein coding genes in LRCA (n=1432) so that the comparison with LhRAtCA and LhRCA can be easily grasped.

Reviewer #3 (Remarks to the Author):

I enjoyed reading this manuscript, which gave an in depth look into the early evolution of the Rickettsiales. As noted, it is an important bacterial clade, giving rise to mitochondria, and Wolbachia, and other major symbionts and pathogens in animals and protozoans. The well known classic members of the group have highly reduced, AT biased genomes and reduced metabolism, reflecting host dependence and intracellular life. Not much was known about the early stages of emergence, from marine lineages. I was interested to see that using some newly retrieved MAGS from marine metagenomic datasets, some of the early steps in the emergence of classic Rickettsiales could be inferred.

The study used many different informatics approaches, including extensive phylogenetics analyses, reconstruction of ancestral gene repertoires. inference of life style given gene repertoires. I went over these to the best of my knowledge, though some details were not familiar to me. In general, the analyses appear solid, and up-to-date, and assumptions are explained.

A minor issue involved the cutoff in ALE for determining gene presence at a node. The cutoff infers presence if it occurs in >0.3 of instances, and the same threshold was used for inferring gene losses and gains. Since the gene repertoires at ancestral nodes are key to the reconstruction of stages of emergence of modern lineages, this seems critical, and I would appreciate seeing more justification for this. I do realize that incompleteness of MAGs and other artifacts will cause nodes to drop genes when they are really present, but some further explanation would be helpful.

In reconstructing the emergence of different stages of host association, the scenarios seem to rely on choosing between intracellular and extracellular host association. But many bacteria are facultatively intracellular. In fact, obligate intracellular symbionts including some gammaproteobacterial ones in insects likely emerged from pathogens that invade eukaryotic cells, such as *Salmonella* using a T3SS. Indeed it seems most likely that an intermediate stage would initially invade cells as part of the life cycle, rather than permanently.

In general, most of these inferences are necessarily speculative. They seem quite reasonable to me. In particular, I agree that reversion from obligate intracellular to extracellular is quite unlikely (though reversion from facultative intracellular to obligate extracellular seems very possible and could be considered here). The general finding that pathways were lost independently in different lineages as they evolved close host associations seems solid, and is of course quite plausible, since this is seen in general in symbiotic/pathogenic bacterial lineages.

Author Rebuttal to Initial comments

Reviewer #1 (Remarks to the Author):

Fantastic job! My comments are mostly minor and meant to provide some resolution where it may be appreciated, and suggestions on clarity. This is a massive effort and will have a strong impact on Rickettsiology. Kudos!

The manuscript titled “The evolutionary origin of host association in an ancient bacterial clade” by Schön and colleagues reports on the discovery of novel lineages of Order Rickettsiales by 1) assembling 11 new genomes from metagenomic datasets obtained from aquatic environments, and 2) conducting robust phylogenomics analyses to define features of the new and existing lineages. Prior to this report, Rickettsiales consisted of three established families (Rickettsiaceae, Anaplasmataceae, Midichloriaceae) and a recently proposed family (Deianiraeaceae) that forms a very long branch in estimated phylogenies. The Rickettsiaceae, Anaplasmataceae, Midichloriaceae comprise obligate intracellular species, while the Deianiraeaceae was predicted to comprise species with a recent transition from facultative to obligate intracellular. Remarkably, the 11 new assembled genomes substantially added to rickettsial diversity such that two basal lineages were revealed (proposed families Mitibacteraceae and Athabascaceae) as well as a lineage most closely related to Deianiraeaceae (proposed family Gamibacteraceae). Detailed phylogenomics analysis are employed to propose a gradual shift from free-living/biofilm-associated lifestyles in the basal Mitibacteraceae and Athabascaceae species. Specific focus on the Rickettsiales vir homolog

(Rvh) type IV secretion system (T4SS) resulted in a hypothesis that this complex has been repurposed throughout rickettsial evolution as this lifestyle shift occurred.

Aside from the human health, biocontrol, and ecological importance of Rickettsiales, no discussions on the evolution of eukaryotes are possible without considering this ancient proteobacterial lineage. Hypotheses on the origin of the mitochondria, particularly the closest extant lineage to the mitochondrial precursor, and subject to change as advances are made in phylogenetic methodology but also as novel species are revealed that help fill in the gaps between extinct and extant lineages. Thus, this work by Schön and colleagues is a substantial contribution to evolutionary biology and the origin of Eukaryota. It also has a profound impact on Rickettsiology, providing insight on how the different lineages have diversified and evolved

discrete relationships with eukaryotic host cells. These novel assemblies will allow estimation of robust phylogenetic frameworks to decipher the evolution of traits underpinning endosymbiosis, reproductive parasitism, and vertebrate virulence (among others). Furthermore, the methodological approach by Schön and colleagues will illuminate the power of metagenomic datasets (if analyzed with care and caution) for illuminating unrealized diversity and advancing classification for virtually any microbial system. Thus, I predict that this work by Schön and colleagues will make a substantial and lasting impact on the fields of evolutionary biology, Rickettsiology, and genomics/phylogenomics.

The manuscript is very well written and presented in just the right format to keep the salient points at the front and important minutia in the supporting materials. Additionally, the authors make their code and scripts available at github. The manuscript is the right length for Nature Microbiology and should have a broad appeal for readers. Below are mostly minor comments (moving right through the manuscript, so supporting information is discussed as it appears throughout).

Response: We would like to thank Reviewer #1 for the positive and constructive feedback, and the in-depth review of our manuscript.

Line 43-44: “leading” used twice, stylistic suggestion.

Response: We have changed this in the revised manuscript. The sentence now reads “Obligate host-associated bacteria include pathogens that represent a leading cause of human, livestock and crop disease, resulting in significant economic loss worldwide”

Line 54 or 56: Consider citing PMID: 28951473 for metabolomic reconstruction.

Response: We have included this citation in the revised manuscript.

Line 66: “In-depth”.

Response: We have corrected this in the revised manuscript.

Line 70: I realize pathogenicity is important but so are other host-relationships that evolved, like mutualism and reproductive parasitism.

Response: We agree, and adjusted the revised text to reflect this as follows: “Subsequent ancestral genome content analysis across Rickettsiales provides new insights about the emergence of host association, a key step in the evolution of various host-relationships pathogenicity indisplayed by this bacterial clade, including pathogenicity, mutualism and reproductive parasitism.”Line 80: Supp. Figure 1 is very difficult to read. I think the font size should be greatly expanded.

Response: The font size is now increased to increase readability..

Line 83: Supp. Figure 2 is also very difficult to read. Aside from small font size, the color scheme might be a bit too much. Maybe symbols and boxes around clades to help with the taxonomic assignments?

Response: The number of different colors has been reduced and shaded boxes have been added to denote particular clades. The font size was kept the same as increasing this would cause overlapping leaf annotations.

Line 101-114: Regarding the shortened branch for *Ca. Deianiraea vastatrix*, have you considered removing this taxon altogether and estimating phylogenies on the datasets with the various employed approaches? Has there been an evaluation of the *Ca. Deianiraea vastatrix* genome to determine its quality? Is it possible it is a mosaic assembly? Please consider citing PMID: 23475938 since it showed a similar lack of corroboration for the placement of *Midichloriaceae* relative to *Rickettsiaceae* and *Anaplasmataceae* (different datasets yielded different, albeit weakly supported, topologies). Perhaps there is still some large swath of diversity we are missing from this region of the tree?

Response: Indeed, the long branch of *Ca. Deianiraea vastatrix* could potentially induce long-branch attraction artefacts. However, we have not observed any major topological shifts in the species tree upon removal of *Deianiraea*. This seems to suggest that the long branch of *Ca. Deianiraea vastatrix* did not cause any phylogenetic artefacts. We have included this analysis in the revised manuscript (methods section and Supplementary Figures 6 & 7).

To our knowledge, there has not been an independent evaluation of the *Ca. Deianiraea vastatrix* genome assembly apart from the original publication, which involved several quality checks such as targeted PCR to close contig gaps. However, in the single gene tree reconstructions of the marker genes used for our phylogenomic reconstructions, *Ca. Deianiraea vastatrix* sequences clustered, as expected, within *Rickettsiales*, indicating that there is no severe non-*Rickettsiales* contamination in the assembly. We already cite the suggested article (PMID: 23475938) in the supplementary text when discussing the different

topologies in more detail. Finally, we fully agree that there is likely a whole group of organisms related to *Ca. Deianiraea vastatrix* and Gamibacteraceae left undiscovered. This however is not the focus of our current manuscript.

Line 119: Why would rDNA sequences not be available? This seems strange that so much other DNA would be present but not rDNA sequences. Are they partial? Are there others available from those metagenomes?Response: This is a result of a common artifact in the assembly and binning process of metagenomic contigs. Ribosomal RNA genes may be repeated several times in the genome and are highly conserved at the nucleotide level and may thus be mis-assembled with rRNA genes of close relatives in the same metagenome. In addition, binning tools assume that all contigs of the same organism have similar read coverage and similar nucleotide compositions. (Short) contigs that contain rRNA genes typically violate these assumptions because rRNA genes display deviant nucleotide compositions compared to the rest of the genome and may have elevated coverage levels due to reads from non-assembled rRNA copies or non-assembled genomes mapping onto these contigs. All these factors lead to the fact that contigs with rRNA do not end up in the same bin as the rest of their genome.

Line 125: The branch support in Supp. Figure 8 is very difficult to read. Also, aren't there many 16S rDNA sequences available for Deianiraeaceae to include?

Response: We increased the font size on all text in the figure (now Supplementary Figure 10). Since the focus of the present manuscript was on the diversity of newly identified families, we did not include additional environmental Deianiraeaceae sequences (even though those are likely available).

Line 125: This is reminiscent of PMID: 23475938 wherein reads were assembled from the Trichoplax genome assembly into a putative symbiont (RETA).

Response: We agree, but in the current context citing this paper (PMID: 23475938) seems less relevant. We however do cite this paper in the supplement when discussing the different possible topologies of the established rickettsial families.

Figure 2: The green shadings for the Athabascaceae and Mitibacteraceae are hard to distinguish. Also, is it really OK to call Deianiraeaceae extracellular? Could it not be an obligate epicellular species? I feel like some readers may be confused by this. I think the references to (b) and (c) are swapped in the legend. While the figure is very informative and powerful in its presentation, it is difficult to read and will hopefully be enlarged for publication. You could decrease the width of the histograms (d-f) to increase the font sizes of all the words.

Response: We adjusted the shading for the Athabascaceae to be more distinct from

Mitibacteraceae. We also agree that Deianiraea should be consistently labeled as 'Ectosymbiont'. We fixed the legend regarding panels b) and c). Finally, we are confident that the editorial team will assist us to achieve optimal visibility/readability of the figure's content.

Supplementary Text: this is a fantastic summary of the phylogenetic methodology and very robust.

Response: Thank you!

Supplementary Text: I see no weakness in the methods or rationale for data exclusion, though the 2-trimming approach could be explained a bit more.

Response: The iterative chi-square trimming approach has now been explained in more detail in the supplementary text, in addition to the description already available in the methods section.

Supplementary Text: In regard to the “Putative T4SS effectors in Mitibacteraceae and Athabascaceae”, PMID: 32546622, PMID: 29946049 and PMID: 26291822 should be cited for PK3, ANK and ARF-GEF domains (eukaryotic) in Rickettsia T4SS effectors.

Response: We now added the mentioned references for Rickettsia T4SS effectors in the revised Supplementary Text.

Supplementary Text: In regard to Supp. Fig. 11, the sequences (accession numbers) from the HMMER searches would be nice as a supporting file to help researchers focus on possible new T4SS effectors. I am sure the results are repeatable, so even the exact query sequence used to get the results shown in the table would be great. If multiple queries were used per category it would be better to just provide a table. I found it difficult to navigate between Supp. Data 5 and Supp. Data 6, and while it is cool to map the distribution of these domains, providing the Uniprot/NCBI accession numbers for each sequence is more helpful for the community.

Response: We provide the Pfam clan/family IDs that correspond to the hidden Markov models that were used to search the genomes in the methods. The results of these searches (which list the sequence IDs of the potential effector proteins) are provided in the figshare repository. Everything that is needed to replicate the analyses we performed to obtain our results is thereby available, either in the Supplementary Information, or in the Figshare repository.

We are unsure what the reviewer means in reference to Supp data 5 and 6 in the context of domain analysis, as these data files contain data from the ancestral gene content (ALE) analyses. Supp Data 6 is simply a summary of the numbers detailed in Supp Data 5. While these tables show the number of copies (as well as duplications etc.) for all nodes including the reconstructed ancestors, the sequence IDs for each cluster and each of the extant taxa

can be found in the annotation table in the FigShare repository ("all protein annotations").

Line 156: Extended Data Figure 1: this is a nice depiction of the metabolic reconstruction; however, the colors are very difficult to assign to the taxa. If this has to remain this small, I suggest an example of the circles/squares with dotted lines connecting the taxa to eachball/square. The lines connection various biochemical processes are difficult to interpret in some cases. What is ALE in the legend?

Response: To increase the accessibility of the figure we added a second legend box that indicates the family or ancestor position and color in our overview taxonomy representations as suggested by the reviewer. We also increased the line width to improve visibility of lines in general.

Line 156: Supp. Data 4: this table is nicely made, but can an NCBI/Uniprot accession be taken from the table? Can a researcher find the protein for a specific taxon easily in this dataset? Again, exemplar sequences (IDs) at least would be helpful. What are the IDs in column A? The legend does not provide enough information.

Response: Thanks for pointing this out. We noted that there has been an unfortunate mistake in the header of this table, which has now been fixed. Column A corresponds to cluster IDs. For each taxon, the corresponding sequence ID(s) for each cluster can be obtained by looking up the cluster ID in the annotation table provided in the FigShare repository (“all protein annotations”).

Line 164: Do the proteins unique to Athabascaceae and Mitibacteraceae show a vertical descendance pattern from the closest Rickettsiales lineage?

Response: Our inferences based on the ancestral reconstruction method ALE, which takes phylogenetic trees into consideration when reconstructing the history of gene families, show that such proteins, which were inferred to likely have been present in LRCA, are generally monophyletic in Athabascaceae and Mitibacteraceae.

Line 181: Is the flagellar apparatus the same across the Rickettsiales lineages that possess it? Or have the Midichloriaceae and/or Rickettsiaceae versions been acquired via lateral gene transfer?

Response: Indeed, when we prepared a phylogeny of the FlhA subunit of the flagellar apparatus with a methodology similar to the single gene trees of subunits of the T4SS. This tree shows that all Rickettsiales copies, including the novel lineages, are monophyletic. Furthermore, the fact that our ancestral reconstructions (which are based on phylogenetic

trees) infer the flagellar apparatus genes in the indicated ancestral nodes also strongly supports that these evolved vertically in the Midichloriaceae and/or Rickettsiaceae. We also provide this tree in the FigShare repository (“FlhA EggNOG tree”).Figure: Flagellar biosynthesis protein FlhA (COG1298) phylogeny supports vertical inheritance of Rickettsiales flagellar apparatus.

Figure 3: Can you switch the direction of the half circle diagonal to distinguish between partial gene presence vs. partial taxon presence? Or do both cases occur per lineage such that it is more complicated?

Response: Indeed there are several cases where both types of 'partialness' occur, making it difficult to visualize it more specifically. It is possible, however, to check which taxa or lineages lack certain genes from Supplementary Data 4.

Figure 3: I think "CONJ" is misleading as presented. Some genomes just have remnants of ICEs while others have a full conjugation apparatus. Most of the Rickettsiaceae have degraded ICEs.

Response: We thank the reviewer for pointing out the misleading representation of conjugative

T4SSs (CONJ) in Rickettsiaceae. As explained by the reviewer this system is only complete in a few genomes (as can also be seen in Supplementary Data 4) and has therefore been changed to a half circle.

Figure 3: The numbers for amino acids inside the orange circles are difficult to make out. Maybe try a darker orange and white numbers?

Response: Thanks for pointing this out. We now changed the color of the circles and numbers representing the number of amino acid synthesis pathways.

Figure 3: Similar as above: do the genes encoding the chemotaxis proteins, T4P, tad, and cbb3-like oxidases show vertical descent from a common ancestor?

Response: The fact that the single gene trees that we reconstructed for each of these components for Rickettsiales and alphaproteobacteria usually show only moderately long branches and that these genes have been inferred to have been present in the last common ancestor of all Rickettsiales (LCRA), strongly supports a vertical descent within either alphaproteobacteria (such as chemotaxis) or Rickettsiales (e.g. T4P) based on our data. However, we cannot exclude the possibilities of e.g. independent gene gains from closely related species outside of the alphaproteobacteria, which could in principle give rise to similar phylogenetic patterns. In order to verify this, larger in-depth phylogenetic analyses of single genes would be required, including a larger taxonomic sampling across e.g. all of bacteria. Such specific analyses are outside the scope of the present study, which aims to point out a more global picture of Rickettsiales gene content evolution.

Figure 3: In panel B, why is RvhB7 not shown? These genes are there, they are just hard to find. Sequences are shown in Figure 4 of PMID: 20176788. I'm sure an HMMER search would find them. P-T4SSs need a lipoprotein in the core complex.

Response: Thanks for pointing this out. We performed a new search specifically for rvhB7 in the selected genomes. Indeed, using a tblastn search we could identify good candidates for rvhB7 in all of them. They were in most cases not predicted by the initial gene calling, and therefore missed in our analyses. We now include them in this figure, and uploaded the candidate sequences to the FigShare repository.

Line 185: It is very difficult to read the text in this figure. Please expand so the font size is legible.

Response: As proposed by the reviewer we have increased the font size in Supplementary Figure 9 (now Supplementary Figure 11).

Line 189: Replace “could” with “can”; as is, it sounds like these bacteria are extinct.

Response: We agree with the reviewer and have implemented the suggested change.

Line 192: Supp. Figure 10 is completely unreadable. Can you make the taxon names larger to the left and right of the tree? There appear to be some red sequences outside of the major RvhB4 duplication. What are these? Also, Figure 2 in PMID: 26646013 showed ancient duplication of the rvh T4SS within a larger T4SS phylogeny.Response: We completely agree that this tree (now Supplementary Figure 12) is very big and was hardly readable. We attempted to improve the readability by splitting it and displaying the two parts side by side as well as improving readability of taxon names and increasing font sizes. Furthermore, we also provide a version of this tree in nexus format in the FigShare repository, so it should be possible to read it like that as well. The sequences marked in red are all taxa from our phylogenomics dataset, including the outgroup.

As the figure in PMID: 26646013 shows as well, rvhB4, rvhB8 and rvhB9 seem to share an ancient duplication, which our data here confirms and which we can trace back even earlier in Rickettsiales evolution. In this tree, two additional copies of virB4 from *Ca. Jidaibacter acanthamoeba* and from 'Rickettsiales bacterium Ac37b' are seen, which are likely the result of HGT events.

Line 194: So, "RARP" was named for specific Rickettsia proteins that are secreted (shown experimentally) and carry ankyrin repeats (PMID: 22773786). RARP-1 was shown to be secreted during host infection, and secretion in a heterologous system (*E. coli*) required a SEC signal sequence and the TolC protein of *E. coli*. So RARP-1 was proposed to be secreted via two-step translocation. RARP-2 was also shown to be secreted during infection and its C-terminal "tail" interacted with RvhD4 in a B2H assay (PMID: 29946049). A pull-down using RvhD4 antibody also captured RARP-2 (among many other proteins, some of which are predicted to be T4SS effectors).

Response: We thank the reviewer for clarifying this point to us. We modified the text regarding this issue accordingly as follows: "Besides a few homologs of rickettsial ankyrin repeat protein (RARP) RARP-1, a Sec-TolC-secreted effector of *Rickettsia typhi*44 and RARP-2, which may be secreted by the T4SS, we were neither able to identify any of the experimentally verified T4SS effector proteins of classical Rickettsiales^{11,12}, nor could we detect similar numbers of putative effector proteins containing eukaryotic-like repeat domains as in other Rickettsiales genomes (ankyrin-, leucine rich- and tetratricopeptide repeats; Supplementary Text; Supplementary Figure 13; Supplementary Data 4)."

Line 197: A newer study characterized Risk-1, a Rickettsia secreted PI3 kinase PMID: 32546622. This protein was also captured in the pull-down described above.

Response: We thank the reviewer for pointing this out to us. We could not identify homologs of Risk-1 (ENOG41025YZ) outside of the Rickettsiaceae and have included the annotation of the Risk-1 gene family in Supplementary data 4.

Line 199: Are the query sequences available (IDs) used in the HMMER searches?

Response: We provide the Pfam clan/family IDs that correspond to the hidden Markov models that were used to search the genomes in the methods (also see above). The results of these searches (which list the sequence IDs of the potential effector proteins) are provided in the Figshare repository ('potential effector domains searches'). Anything that is needed to replicate the analyses we performed is available, either in the Supplementary material or the Figshare repository.

Line 206-216: I agree that the combined observations support Athabascaceae and Mitibacteraceae as facultative or extracellular. Why, then, is the VirB5 protein absent from these T4SSs? Most P-T4SSs elaborate a pilus that needs the VirB5 protein.

Response: This is an excellent question that we can only speculate about. One scenario could be another protein performing the minor pilin function in these lineages, like is the case for CagL in *Helicobacter pylori* (PMID: 21909278), which performs the VirB5 function but is not homologous to it.

Extended Data Figure 2: the numbers inside the blue circles cannot be seen. Try white numbers. Are the various gene gains supported by phylogeny estimation? In other words, do phylogenies indicate LGT? If so, from where?

Response: As proposed by the reviewer, we changed the font color within the circles of Extended Data Figure 2 to white. The various gene gains and losses represent the results from the ALE reconstruction, which indeed is based on phylogenetic reconstruction. However, using this approach, it is usually not possible to distinguish between 'de-novo' genes or LGTs from outside of the sampled taxa, i.e. in this case from outside of alphaproteobacteria.

Line 253: It could also be a conjugation system or involved in DNA uptake. Both the ancestral and modern form. What is fascinating is the oddity of the RvhB4,B8,B9 duplication and expansion of the large RvhB6 proteins. Why are these present across all of the species even when there are now such distinct lifestyles?

Response: We fully agree with the reviewer, this is definitely a possibility. It is still unknown if Rickettsiales produce only one single T4 secretion machinery and what role different paralogs (RvhB4,B8,B9 and the expanded Rvh6) play (PMID: 27307105). While this is a very interesting

point, in the scope of our study we did not dare speculate about alternative functions of the Rickettsiales T4SS outside of a protein secretion system.

Line 266: So, provided that *Ca. D. vastatrix* is still “obligate” from a metabolic perspective, perhaps the P-T4SS is involved in sucking nutrients out of cells?

Response: This could indeed be speculated; In the predatory CPR bacterium *Vampirococcus lugosii*, however, the type 4 pilus (T4P) together with the competence-related integral

membrane protein ComEC has been proposed to be involved in the uptake of host dna (PMID: 33911080). *C. vastatrix* encodes both a (partial) T4P and ComEC, making it a likely candidate system for nutrient uptake.

Line 338: I am not sure what is meant by the rickettsial OMP. Rickettsia have an expanded autotransporter family called Surface Cell Antigens (Scas); Orientia has counterparts. If this is what is being referred to, the references for these should be cited. Using “OMP” is very misleading.

Response: We thank the reviewer for pointing this out. Indeed, we aimed to refer to the family of surface cell antigens here (cluster 023YT) and corrected this in the manuscript.

Figure 4: Some Rickettsia have the ability to convert acetate to acetyl-CoA using AckA and Pta. Not sure if those are the enzymes referred to in the Losses box.

Response: As pointed out by the reviewer, as inferred our reconstruction of the LRCA, some Rickettsia species (e.g. *Rickettsia bellii*) are capable of converting acetate to acetyl-CoA by the concerted action of AckA and Pta. However, our reconstructions infer a loss of these two genes in LhRCA, the ancestor of all classical Rickettsiales, and an independent regaining within the genus Rickettsia.

Line 502: Per my notes above for lacking sequence information: if the Figshare repository has this information, then ignore my comments. I only ask that all figures and tables listing presence/absence of genes allow for repeatability.

Response: All information and data required to reproduce the results shown in the present manuscript are either available as Supplementary data or in the Figshare repository. We now also uploaded the protein sequences from all taxa presented and used in the analyses here. In this case, we provide the results files from the HMMer searches which list all sequences from all genomes analysed here. As HMMs we used the publicly available files for PFAM families/clans listed here.

Line 602: Is it possible to search again for VirB7 and VirB5? I am not sure why this were not included. It would be cool to find VirB5 in the Athabascaceae and Mitibacteraceae.

Response: As described above (and in the revised methods), we searched the selection of genomes presented in Figure 3b for candidate sequences of virB7, using tblastN. We were able to identify good candidates in all genomes, even where they had been missed before (since they were not annotated by prodigal). However, we were unable to find any candidates for virB5 in any of these genomes using either tblastN or HMMER searches.Materials and Methods. Excellent. The workflow is explained well and should be repeatable if all sequence information is accessible. What a massive effort!

We thank the reviewer for their appreciation of our description of the workflow.

Reviewer #2 (Remarks to the Author):

This is a very exciting new paper presenting eleven new metagenome assembled genomes of organisms within the Rickettsiales, placed in three clades. At least two of these are likely to represent organisms that are free living or host associated (but not obligate intracellular). This work adds to a growing body of research that is expanding our view of the Rickettsiales beyond the well described families of Rickettsiaceae and Anaplasmataceae. The recent publication of *Ca. Deianiraea vastatrix* gave the first indication of some free-living Rickettsiales, and the current manuscript greatly strengthens this finding. This work will have a profound impact on our understanding of Rickettsiales and the adaptation to obligate intracellular replication. I am very enthusiastic about this manuscript and have only minor comments:

Response: We would like to thank the reviewer for the feedback and comments. We would however like to point out that as part of the current study we only generated three 'new' MAGs; the other newly analyzed MAGs were made available as part of large scale metagenome binning studies (hence generated by other researchers' efforts).

1. Line 80-83: the use of mitochondrially encoded marker genes is not clear on a first reading and should be elaborated.

Response: We have rephrased this sentence to increase clarity on the nature of the phylogenomics dataset pertaining the mitochondrially encoded genes.

2. Line 197-198: what do they mean by not being able to detect "increased numbers" of putative effector proteins? Increased relative to what? Were Anks/TPRs detected and if so how many?

32Response: We changed the text in order to be more clear and transparent here: “we [could not] detect similarly large numbers of putative effector proteins containing eukaryotic-like repeat domains as in other Rickettsiales genomes”. The number of detected eukaryotic-like repeat domains can be obtained from Supplementary Figure 11 (now Supplementary Figure 13), and in the corresponding section of the supplementary text we also provide the numbers of ANKs/TPRs found for the new lineages.3. Line 245: the discussion of the expanded gene repertoire in *Orientia* should include mention of the very high number of pseudogenes.

Response: We now added a sentence that mentions the high number of pseudogenes in the *Orientia* genome.

4. Line 269-271: It would help the reader to include again the number of predicted protein coding genes in LRCA (n=1432) so that the comparison with LhRAtCA and LhRCA can be easily grasped.

Response: We completely agree with this suggestion and added the number of reconstructed protein coding genes in LRCA to make the comparison easier to grasp.

Reviewer #3 (Remarks to the Author):

I enjoyed reading this manuscript, which gave an in depth look into the early evolution of the Rickettsiales. As noted, it is an important bacterial clade, giving rise to mitochondria, and Wolbachia, and other major symbionts and pathogens in animals and protozoans. The well known classic members of the group have highly reduced, AT biased genomes and reduced metabolism, reflecting host dependence and intracellular life. Not much was known about the early stages of emergence, from marine lineages. I was interested to see that using some newly retrieved MAGS from marine metagenomic datasets, some of the early steps in the emergence of classic Rickettsiales could be inferred.

The study used many different informatics approaches, including extensive phylogenetics analyses, reconstruction of ancestral gene repertoires. inference of life style given gene repertoires. I went over these to the best of my knowledge, though some details were not familiar to me. In general, the analyses appear solid, and up-to-date, and assumptions are explained.

Response: We thank the reviewer for the positive and constructive response and feedback.

A minor issue involved the cutoff in ALE for determining gene presence at a node. The cutoff

34infers presence if it occurs in >0.3 of instances, and the same threshold was used for inferring gene losses and gains. Since the gene repertoires at ancestral nodes are key to the reconstruction of stages of emergence of modern lineages, this seems critical, and I would appreciate seeing more justification for this. I do realize that incompleteness of MAGs and other artifacts will cause nodes to drop genes when they are really present, but some further explanation would be helpful.Response: The number of ‘events’ (duplications, losses, transfers, originations and copies) that are inferred by ALE follow a bimodal distribution, with most numbers being either close to 1 (very strong signal for presence) or close to 0 (very strong signal for absence). The threshold of 0.3 represents a choice for including somewhat weakly supported events, while avoiding the ‘noise’ at the bottom end of the distribution. We chose this permissive threshold in order to be able to for example also reconstruct complete pathways where some genes show a clear signal for presence while others are only weakly supported. When comparing the number of inferred gene families at the thresholds of 0.3 and 0.5 for two important nodes, LRCA and LhRCA, there is only a marginal difference that can be observed (1241 vs. 1121 for LRCA and 1034 vs. 971 for LhRCA), confirming this decision. Finally, we wanted to apply a similar threshold for all types of events in order to be more consistent. Below, we visualize these distributions and we also include this analysis in the revised manuscript (Supplementary Figure 14).

In reconstructing the emergence of different stages of host association, the scenarios seem to rely on choosing between intracellular and extracellular host association. But many bacteria are facultatively intracellular. In fact, obligate intracellular symbionts including some gammaproteobacterial ones in insects likely emerged from pathogens that invade eukaryotic cells, such as *Salmonella* using a T3SS. Indeed it seems most likely that an intermediate stage would initially invade cells as part of the life cycle, rather than permanently.

Response: We completely agree that our inferences would also support a scenario where

intermediate stages of facultative intracellular lifestyles predated the obligate intracellular lifestyles in the early evolutionary history of the Rickettsiales. We have tried to rephrase the corresponding parts of the manuscript to reflect the possibility for a facultative intracellular lifestyle of LhRCA.

In general, most of these inferences are necessarily speculative. They seem quite reasonable to me. In particular, I agree that reversion from obligate intracellular to extracellular is quite

unlikely (though reversion from facultative intracellular to obligate extracellular seems very possible and could be considered here). The general finding that pathways were lost independently in different lineages as they evolved close host associations seems solid, and is of course quite plausible, since this is seen in general in symbiotic/pathogenic bacterial lineages.

Response: Indeed, it is nice to see that the obtained results fit nicely with the most plausible evolutionary scenario (even though evolution does not necessarily always follow 'logics', and should not be dismissed a priori!)

Decision Letter, first revision:

Dear Dr. Ettema,

Thank you for submitting your revised manuscript "The evolutionary origin of host association in an ancient bacterial clade" (NMICROBIOL-22010142A). We considered that the reviewers comments have been satisfactorily addressed, and therefore we'll be happy in principle to publish it in Nature Microbiology, pending minor revisions to comply with our editorial and formatting guidelines.

Thank you again for your interest in Nature Microbiology Please do not hesitate to contact me if you have any questions.

Sincerely,

{redacted}

Author Rebuttal, first revision:

1Decision Letter, final checks:

Dear Thijs,

Thank you for your patience as we've prepared the guidelines for final submission of your Nature Microbiology manuscript, "The evolutionary origin of host association in an ancient bacterial clade" (NMICROBIOL-22010142A). Please carefully follow the step-by-step instructions provided in the attached file, and add a response in each row of the table to indicate the changes that you have made. Please also check and comment on any additional marked-up edits we have proposed within the text. Ensuring that each point is addressed will help to ensure that your revised manuscript can be swiftly handed over to our production team.

In recognition of the time and expertise our reviewers provide to Nature Microbiology's editorial process, we would like to formally acknowledge their contribution to the external peer review of your manuscript entitled "The evolutionary origin of host association in an ancient bacterial clade". For those reviewers who give their assent, we will be publishing their names alongside the published article.

Nature Microbiology offers a Transparent Peer Review option for new original research manuscripts submitted after December 1st, 2019. As part of this initiative, we encourage our authors to support increased transparency into the peer review process by agreeing to have the reviewer comments, author rebuttal letters, and editorial decision letters published as a Supplementary item. When you submit your final files please clearly state in your cover letter whether or not you would like to participate in this initiative. Please note that failure to state your preference will result in delays in accepting your manuscript for publication.

Cover suggestions

As you prepare your final files we encourage you to consider whether you have any images or illustrations that may be appropriate for use on the cover of Nature Microbiology.

2Covers should be both aesthetically appealing and scientifically relevant, and should be supplied at the best quality available. Due to the prominence of these images, we do not generally select images featuring faces, children, text, graphs, schematic drawings, or collages on our covers.

Nature Microbiology has now transitioned to a unified Rights Collection system which will allow our Author Services team to quickly and easily collect the rights and permissions required to publish your work. Approximately 10 days after your paper is formally accepted, you will receive an email in providing you with a link to complete the grant of rights. If your paper is eligible for Open Access, our Author Services team will also be in touch regarding any additional information that may be required to arrange payment for your article.

Please note that *Nature Microbiology* is a Transformative Journal (TJ). Authors may publish their research with us through the traditional subscription access route or make their paper immediately open access through payment of an article-processing charge (APC). Authors will not be required to make a final decision about access to their article until it has been accepted. [Find out more about Transformative Journals](https://www.springernature.com/gp/open-research/transformative-journals)

Authors may need to take specific actions to achieve [compliance with funder and institutional open access mandates](https://www.springernature.com/gp/open-research/funding/policy-compliance-faqs). If your research is supported by a funder that requires immediate open access (e.g. according to [Plan S principles](https://www.springernature.com/gp/open-research/plan-s-compliance)) then you should select the gold OA route, and we will direct you to the compliant route where possible. For authors selecting the subscription publication route, the journal's standard licensing terms will need to be accepted, including [self-archiving policies](https://www.nature.com/nature-portfolio/editorial-policies/self-archiving-and-license-to-publish). Those licensing terms will supersede any other terms that the author or any third party may assert apply to any version of the manuscript.

For information regarding our different publishing models please see our page

[href="https://www.springernature.com/gp/open-research/transformative-journals">](https://www.springernature.com/gp/open-research/transformative-journals) Transformative Journals page. If you have any questions about costs, Open Access requirements, or our legal forms, please contact ASJournals@springernature.com.

Please use the following link for uploading these materials:
{redacted}

Best regards,

{redacted}

Final Decision Letter:

Dear Professor Ettema,

I am pleased to accept your Article "The evolutionary origin of host association in the Rickettsiales" for publication in Nature Microbiology. Thank you for having chosen to submit your work to us and many congratulations.

After the grant of rights is completed, you will receive a link to your electronic proof via email with a request to make any corrections within 48 hours. If, when you receive your proof, you cannot meet this deadline, please inform us at rjsproduction@springernature.com immediately. You will not receive your proofs until the publishing agreement has been received through our system.

Acceptance of your manuscript is conditional on all authors' agreement with our publication policies (see <https://www.nature.com/nmicrobiol/editorial-policies>). In particular your manuscript must not be published elsewhere and there must be no announcement of the work to any media outlet until the publication date (the day on which it is uploaded onto our website).

4Please note that *Nature Microbiology* is a Transformative Journal (TJ). Authors may publish their research with us through the traditional subscription access route or make their paper immediately open access through payment of an article-processing charge (APC). Authors will not be required to make a final decision about access to their article until it has been accepted. [Find out more about Transformative Journals](https://www.springernature.com/gp/open-research/transformative-journals)

Authors may need to take specific actions to achieve [compliance with funder and institutional open access mandates](https://www.springernature.com/gp/open-research/funding/policy-compliance-faqs). If your research is supported by a funder that requires immediate open access (e.g. according to [Plan S principles](https://www.springernature.com/gp/open-research/plan-s-compliance)) then you should select the gold OA route, and we will direct you to the compliant route where possible. For authors selecting the subscription publication route, the journal's standard licensing terms will need to be accepted, including [self-archiving policies](https://www.nature.com/nature-portfolio/editorial-policies/self-archiving-and-license-to-publish). Those licensing terms will supersede any other terms that the author or any third party may assert apply to any version of the manuscript.

To assist our authors in disseminating their research to the broader community, our SharedIt initiative provides you with a unique shareable link that will allow anyone (with or without a subscription) to

5read the published article. Recipients of the link with a subscription will also be able to download and print the PDF.
